# VEM: Environment-Free Exploration for Training GUI Agent with Value Environment Model

## Abstract

Training Vision-Language Models (VLMs) for Graphical User Interfaces (GUI) agents via Reinforcement Learning (RL) faces critical challenges: environment-based RL requires costly interactions, while environment-free methods struggle with distribution shift and reward generalization. We propose an environment-free RL framework that decouples value estimation from policy optimization by leveraging a pretrained Value Environment Model (VEM), which requires no live environment interaction during policy optimization. VEM predicts state-action values directly from offline data, distilling human-like priors about GUI interaction outcomes without requiring next-state prediction or environmental feedback. This avoids compounding errors and enhances resilience to UI changes by focusing on semantic reasoning (e.g., "Does this action advance the user's goal?"). The framework operates in two stages: (1) pretraining VEM to estimate long-term action utilities and (2) guiding policy exploration with frozen VEM signals, enabling layout-agnostic GUI automation. Evaluated across diverse benchmarks including Android-in-the-Wild for mobile apps and Multimodal-Mind2Web for web environments, VEM achieves state-of-the-art or highly competitive performance in both offline and online settings. It significantly outperforms environment-free baselines and matches or exceeds environment-based approaches, crucially without incurring interaction costs. Importantly, VEM demonstrates that robust, generalizable GUI agents can be trained efficiently using semantic-aware value estimation, proving effective across distinct interaction platforms like mobile and web. The code is available at https://anonymous.4open.science/r/VEM-Agent-51E7.

## 1 Introduction

Vision-Language Models (VLMs) have shown strong capabilities in common-sense reasoning, abstraction, and generalization, enabling applications across domains Zhou et al. (2022); Zhang et al. (2024d), including autonomous GUI agents Zhang et al. (2024a); Wang et al. (2024b); Nguyen et al. (2024). These agents leverage VLMs to interpret visual and textual inputs for automating GUI interactions, such as locating and clicking interface elements based on natural language commands. This allows automation of tasks ranging from web navigation to complex software operations Yan et al. (2023); Zhang & Zhang (2023); Zhang et al. (2023); Rawles et al. (2024); Bai et al. (2024); Hong et al. (2024). Despite the progress in general-purpose models like GPT-4o OpenAI (2024) and Gemini 1.5 Pro DeepMind (2024), real-world GUI tasks remain challenging due to the variability of interfaces. Minor layout changes (e.g., pop-ups or repositioned buttons) can lead to misinterpretation Xie et al. (2024); Zhang et al. (2024b;e); Bai et al. (2024); Zhang et al. (2024a). This highlights the need for specialized VLMs optimized for GUI tasks. Reinforcement Learning (RL) Sutton (2018) is an effective approach for aligning models with target behaviors Zhai et al. (2024); Sun et al. (2024). In *environment-based RL*, models learn by interacting with environments and receiving feedback Toyama et al. (2021); Bai et al. (2024); Carta et al. (2023); Wang et al. (2024c); Lai et al. (2024), but these methods suffer from high interaction costs and sample inefficiency Xie et al. (2021); Niu et al. (2022). Simulated environments Chae et al. (2024); Gu et al. (2024) can alleviate this but often face fidelity issues and compounding prediction errors Guan et al. (2023); Zhang et al. (2024f); Ge et al. (2024). Alternatively, *environment-free RL* leverages offline RL Snell et al. (2022); Hong et al. (2023); Bai et al. (2024); Wang et al. (2024a) or reward model training Stiennon et al. (2020); Ouyang et al. (2022), avoiding the need for online interaction. However, offline RL is challenged by

distribution shift Levine et al. (2020) and limited exploration Prudencio et al. (2023), while reward model-based methods may fail to generalize in dynamic GUI scenarios where interface changes render static reward signals ineffective Stiennon et al. (2020).

As shown in Figure 1, humans excel at estimating the long-term utility of actions (i.e., state-action values $Q(s, a)$) without explicitly seeing the next state. VLMs, trained on vast corpora of human-GUI interactions, inherently encode similar priors about action outcomes Hao et al. (2023); Bai et al. (2023); Chen et al. (2023). To operationalize this capability and address aforementioned challenges, we propose an environment-free RL framework that decouples value estimation from policy optimization. Unlike prior approaches that rely on reward models or require direct interaction with the environment, our method leverages a pretrained value environment model (VEM) to directly approximate state-action values from offline data. A key distinction here lies in feedback and value estimation: traditional reward models typically provide sparse, trajectory-level feedback (i.e., evaluating an entire sequence of actions rather than individual steps), whereas the VEM learns a dense, step-wise value function—one that estimates the long-term utility of each individual action in the sequence. This dense, step-specific value signal delivers more granular and action-

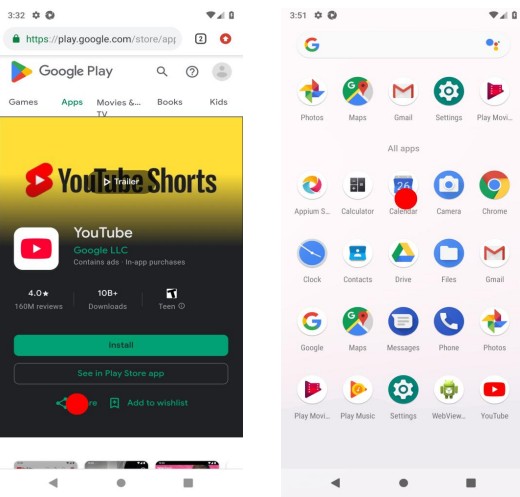

(a) Task: What's the latest video from GameSpot Reviews?  (b) Task: Open the calendar and show me this week's events?

Figure 1: Two GUI tasks with the action marked as a red dot (🔴). In (a), clicking the 'share' button is unlikely to reveal reviews. In (b), the action may open the calendar app to display weekly events.

able guidance, which is particularly critical for complex multi-step tasks where assigning credit to specific actions is otherwise difficult. By distilling human-like priors into a frozen VEM, our policy model bypasses the need for explicit reward engineering or error-prone next-state simulations. Additionally, the VEM's resilience to superficial UI changes stems from its focus on semantic reasoning: it evaluates an action based on its contribution to the overall task goal (e.g., "Does this action advance the user's goal?"), rather than attempting to predict brittle, pixel-level next states.

Concretely, our framework operates in two stages:

**1. Value Environment Model Pretraining:** The VEM is trained offline to predict state-action values $Q(s, a)$, capturing the long-term utility of actions in diverse GUI contexts. This avoids the compounding errors of next-state prediction by focusing on value estimation, which aligns better with VLMs' inherent reasoning strengths.

**2. Policy Exploration with Frozen VEM:** During policy training, the VEM provides value-guided signals to iteratively refine the policy's action selection. By directly exploring for high-value actions that are grounded in the VEM's understanding of GUI semantics, the policy learns to generalize across unseen layouts and functionalities without online interaction.

To evaluate our method, we conduct rigorous experiments across diverse GUI automation benchmarks, specifically Android-in-the-Wild (AITW) Rawles et al. (2024) for mobile applications and Multimodal-Mind2Web (MM-Mind2Web) Deng et al. (2023) for web-based tasks, using dual offline/online protocols. Across both platforms, our approach achieves strong performance. On the AITW benchmark, with only 500 training trajectories (equivalent to one-third the scale of the DigiRL/DigiQ Bai et al. (2024; 2025a) dataset), our approach achieves 34%/35% offline task success rates on General/Webshopping domains, outperforming environment-free counterparts by 5-26% and exceeding previous environment-based methods by 8%. In AITW online deployment, we attain 49.15% general task success, surpassing environment-free methods by 23-49% while remaining comparable to environment-based policies (38.98%) in procedural efficiency (7.59 vs. 7.25 average steps). On the MM-Mind2Web dataset, our method achieves success rates of 55.8%/51.2%/50.5% on the cross-task/cross-website/cross-domain test sets, outperforming other methods trained on the same MM-Mind2Web training set by 4-35%. Notably, we only use 7.7k training samples, yet our per-

formance surpasses models trained on large-scale pretraining data by 2-39%. Crucially, our method eliminates catastrophic failures seen in generic models (e.g., GPT-4o) and achieves substantial relative improvements over the strongest baselines across both mobile and web environments, all without environmental interaction costs. These results demonstrate that offline policy optimization guided by VEM can rival online-trained systems while significantly advancing environment-free paradigms for broader GUI automation tasks.

## 2 RELATED WORKS

### 2.1 ENVIRONMENT-BASED METHODS

Environment-based RL methods train VLMs through direct interaction with GUI environments, where rewards are explicitly provided by the environment. Several frameworks like AndroidEnv Toyama et al. (2021) and DistRL Wang et al. (2024c) enable agents to learn through trial-and-error interactions with real-world digital interfaces. Recent works such as DigiRL Bai et al. (2024) and AutoWebGLM Lai et al. (2024) demonstrate that environment-based RL can effectively align VLMs with complex GUI navigation tasks through autonomous exploration. However, these methods face significant limitations in sample efficiency due to the high cost of environment interactions Xie et al. (2021); Niu et al. (2022), particularly in real-world applications where collecting online feedback is expensive and suffers from high latency. To address this, some approaches like WebRL Qi et al. (2024) and WorldGPT Ge et al. (2024) attempt to simulate GUI environments, but they struggle with state prediction accuracy and compounding errors in long interaction sequences Guan et al. (2023); Zhang et al. (2024f). The fundamental challenge lies in the dynamic nature of real-world GUIs, where interface elements and layouts frequently change, making environment-dependent reward signals inherently unstable Zhang et al. (2024b;e).

### 2.2 ENVIRONMENT-FREE METHODS

Traditional planning-based methods, such as Agent Q Putta et al. (2024), ReST-MCTS Zhang et al. (2024c), and QLASS Lin et al. (2025), rely on Monte Carlo Tree Search or similar algorithms. These approaches require extensive environment interaction and on-policy data, limiting their practicality in real-world GUI scenarios due to privacy, latency, and cost issues. Environment-free methods address these limitations by using offline datasets or learned reward models. Offline RL techniques Snell et al. (2022); Hong et al. (2023) fine-tune vision-language models (VLMs) with pre-collected trajectories, as seen in large-scale GUI agent training Wang et al. (2024a); Bai et al. (2024). Reward modeling Stiennon et al. (2020); Ouyang et al. (2022) predicts task success from static datasets, enabling behavior alignment without real-time feedback. General-purpose VLMs like GPT-4o OpenAI (2024) leverage pre-trained capabilities for zero-shot GUI understanding Yan et al. (2023); Zhang et al. (2023), offering out-of-the-box solutions without RL training. However, environment-free methods face challenges. Offline RL suffers from distribution shift in novel GUI configurations Levine et al. (2020), and reward models are sensitive to interface changes Prudencio et al. (2023). Advanced VLMs like GPT-4o struggle with GUI complexity due to insufficient task-specific fine-tuning Xie et al. (2024); Zhang et al. (2024b). Recent hybrid approaches aim to bridge these gaps. SeeClick Cheng et al. (2024) combines environment-free pretraining with targeted fine-tuning, while Digi-Q Bai et al. (2025a) learns a Q-value function for action generation without full environment simulation. Our method introduces a Value Environment Model (VEM) that learns environment dynamics and value structure offline, reducing reliance on real-environment rollouts for scalable GUI agent development.

## 3 METHOD

This section presents our method for training a GUI agent with offline data, followed by an extended theoretical analysis. As shown in Figure 2, we first describe how to learn VEM from GPT-4o labeled data, then show that the resulting policy can achieve near-optimal performance under certain coverage and accuracy conditions. We further incorporate distribution-shift arguments to highlight the relationship between dataset quality and final policy performance.

## 3.1 PRELIMINARY

We formalize GUI navigation as a Markov Decision Process $\mathcal{M} = (\mathcal{S}, \mathcal{A}, P, r, \gamma)$ where: States $s \in \mathcal{S}$: Task descriptions + interaction histories + current GUI screenshot. Actions $a \in \mathcal{A}$: Our Agent Action design is compatible with the corresponding benchmark, as detailed in Appendix C. $P(s'|s, a)$: Unknown environment dynamics. Rewards $r(s, a) \in \{1, 2\}$: 1 for suboptimal actions, 2 for optimal actions. $\gamma \in [0, 1)$: Discount factor.

Given an *offline* dataset $\mathcal{D} = \{(s_i, a_i, r_i, s'_i)\}_{i=1}^N$ collected by unknown behavior policies or suboptimal behavior policy $\beta$, our goal is to learn a policy $\pi_\phi(a|s)$ maximizing expected returns without environment interaction.

## 3.2 VALUE ENVIRONMENT MODEL TRAINING

A core challenge in GUI automation is the scarcity of explicit reward signals that indicate whether a chosen action advances or hinders task completion. While the benchmark datasets include trajectory-level success labels, they lack the more granular, step-level value supervision necessary for training a Q-function. Therefore, we leverage GPT-4o to generate these necessary dense value signals based on the full trajectory context. Our strategy is to generate coarse but direct supervision by leveraging GPT-4o's understanding of the GUI context. Specifically, we assign each state-action pair a binary label, capturing whether the action is beneficial or detrimental to the target task. This annotated supervision then guides the learning of a VEM, which is a state-action value function $Q_\theta$, avoiding the need for explicit environment interactions.

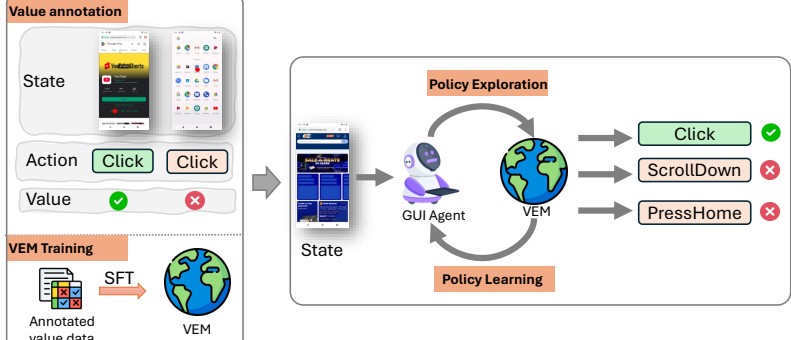

Figure 2: VEM Architecture: (1) Offline dataset annotation using GPT-4o's task understanding, and VEM training via supervised regression. (2) Policy optimization through frozen VEM maximization, encouraging the policy model to explore high-value actions.

**LLM-Guided Annotation.** For each state-action pair $(s, a)$ in our offline dataset $\mathcal{D}$, we leverage GPT-4o with chain-of-thought reasoning Wei et al. (2022) to generate binary labels $\ell(s, a) \in \{1, 2\}$, simulating human-like task progress assessment. This choice simplifies the annotation process, and the specific values do not affect training as they are normalized before optimization (see Appendix E for details). By providing the LLM with the full task and trajectory context, it assesses an action's semantic contribution to the final goal. By conditioning on the full interaction context rather than predicting explicit next states, VEM learns to judge whether an action semantically advances the task goal. This encourages reasoning that abstracts away from surface-level layout details and emphasizes task-level semantics. These annotations provide coarse supervision signals, where $\ell = 2$ indicates actions expected to aid task completion, while $\ell = 1$ marks potentially counterproductive steps. This labeling scheme approximates long-term value through immediate assessments, circumventing the need for explicit reward engineering.

**Supervised Value Learning**. Using the annotated subset $\widetilde{\mathcal{D}} = \{(s_i, a_i, \ell_i)\}$, we fine-tune a Qwen2.5VL Bai et al. (2025b) model to predict label values through mean squared error minimization:

$$\min_\theta \mathbb{E}_{(s,a,\ell)\sim\widetilde{\mathcal{D}}} \left[ (Q_\theta(s, a) - \ell)^2 \right]$$

The trained $Q_\theta$ estimates action quality through environmental understanding distilled from GPT-4o's annotations, rather than online interactions, to reduce cost and latency during subsequent training.

**Stable Policy Guidance**. After convergence, we freeze $Q_\theta$ as a fixed VEM to provide consistent action evaluations. While the binary labels represent simplified supervision, they effectively encode task progression patterns that guide subsequent policy learning. This approach maintains stability by decoupling environment modeling from policy optimization, while remaining fully offline-trainable.

### 3.3 POLICY LEARNING WITH THE FROZEN VEM

Having established a VEM that can evaluate actions in any given GUI state, we now wish to derive a policy that selects actions maximizing the predicted value. By freezing $Q_\theta$ as a fixed state-action value estimator, we transform policy learning into a stable optimization problem that leverages consistent value predictions without environment interactions.

### 3.4 VALUE MAXIMIZATION WITH FROZEN VALUE MODEL

We formulate policy learning as a value maximization problem guided by a *frozen* Q-function $Q_\theta(s, a)$, referred to as the **Value Environment Model (VEM)**. This model is pre-trained offline using trajectory-level annotations from GPT-4o and remains unchanged during policy optimization.

The policy $\pi_\phi(a \mid s)$ is optimized to maximize the expected value estimated by the fixed VEM over the offline dataset $\mathcal{D}$:

$$\max_\phi \ \mathbb{E}_{s\sim\mathcal{D},\, a\sim\pi_\phi(\cdot|s)} \left[ \underbrace{Q_\theta(s, a)}_{\text{frozen}} \right].$$

This frozen Q-function serves as a surrogate reward signal, allowing policy optimization without further environment interaction. The decoupling of value learning and policy learning ensures training stability and computational efficiency.

**Coverage Regularization via SFT Initialization.** To reduce the risk of distribution shift and ensure that the policy remains within the support of the offline dataset, we initialize $\pi_\phi$ using supervised fine-tuning (SFT) on behavior trajectories. This anchors the policy close to the behavior policy $\beta$, implicitly encouraging good coverage of $\mathcal{D}$ without requiring explicit regularization during policy optimization.

**Policy Optimization.** We apply standard policy gradient methods to optimize $\pi_\phi$ against the fixed Q-function. The gradient of the objective is:

$$\nabla_\phi \mathcal{J}(\pi_\phi) = \mathbb{E}_{s\sim\mathcal{D},\, a\sim\pi_\phi(\cdot|s)} \left[ \nabla_\phi \log \pi_\phi(a \mid s) \cdot Q_\theta(s, a) \right],$$

where $Q_\theta$ remains fixed throughout. This update increases the likelihood of high-value actions, leveraging static supervision from the VEM.

**Distinction from Traditional RL.** Unlike actor–critic or Q-learning frameworks—which jointly update the value and policy networks—our method explicitly separates these stages. $Q_\theta$ is trained once offline, then frozen during RL. This separation offers the following advantages: (1) No environment interactions are needed after Q-learning. (2) A fixed target avoids instability from bootstrapping errors. (3) Policy learning reduces to maximizing a fixed, pretrained signal.

**Interpretation of the Frozen Value Model.** The VEM $Q_\theta$ encodes action quality based on task-level success signals, without depending on transition dynamics. It replaces traditional reward feedback, transforming policy learning into a form of offline, value-guided behavior cloning. Training over static data with a frozen value model yields stable updates and avoids the variance associated with on-policy rollouts or dynamic targets. This is especially suited for GUI domains, where environment resets are expensive or unavailable.

## 3.5 THEORETICAL ANALYSIS

We now present a formal performance guarantee under realistic offline RL conditions. Specifically, we show that if the learned value model $Q_\theta$—trained offline and *frozen* during policy optimization—approximates the true optimal Q-function $Q^*$ on the support of the offline dataset $\mathcal{D}$, and the learned policy remains sufficiently close to the data distribution, then the resulting policy achieves provably bounded suboptimality.

**Coverage and Approximate Q-Function Assumptions**   Let $d_\pi(s, a)$ denote the state–action visitation distribution of policy $\pi$. We define two key conditions for our analysis:

*(1) Coverage Ratio.* $\mathrm{Cov}(\mathcal{D}, \pi) = \mathbb{E}_{(s,a) \sim d_\pi} \left[ \mathbf{1}\{(s, a) \in \mathrm{supp}(\mathcal{D})\} \right].$ We require $\mathrm{Cov}(\mathcal{D}, \widehat{\pi}) \geq 1 - \delta$, meaning the learned policy rarely selects actions outside the dataset's support.

*(2) Q-Function Approximation.* We assume $Q_\theta$ is a uniformly accurate approximation of $Q^*$ over the dataset support: $\left| Q_\theta(s, a) - Q^*(s, a) \right| \leq \varepsilon \quad \forall (s, a) \in \mathcal{D}.$

**Performance Bound with Frozen Value Model**   The following theorem quantifies the suboptimality of the policy $\widehat{\pi}$, which maximizes the frozen Q-values over offline states:

**Theorem 3.1** (Frozen Q Performance Bound). *Let* $\widehat{\pi} = \arg\max_\pi \ \mathbb{E}_{s \sim \mathcal{D}, a \sim \pi(\cdot|s)} \left[ Q_\theta(s, a) \right],$ *and suppose that the coverage and approximation assumptions above hold. Then, there exists a constant $c > 0$ (depending on the discount factor $\gamma$ and episode horizon) such that:*

$$J(\pi^*) - J(\widehat{\pi}) \leq c \cdot (\varepsilon + \delta),$$

*where $\pi^*$ denotes the optimal policy.*

The proof follows from standard performance difference bounds and offline RL distribution mismatch arguments (see Appendix D for details). It highlights that the total performance gap is bounded by the Q-function error $\varepsilon$ and the dataset coverage gap $\delta$.

**Assumption Validity and Practical Considerations**   In our implementation, the VEM $Q_\theta$ is trained using GPT-4o-generated value labels that reflect task completion success. Human validation on 50 random samples from each dataset shows a 90% agreement rate (see details in Appendix I), suggesting that the approximation error $\varepsilon$ is small in practice. To control coverage shift, we initialize $\pi_\phi$ via supervised fine-tuning (SFT) on behavior trajectories. This anchors the policy distribution near the dataset's support without requiring an explicit KL penalty during policy optimization. Empirically, we measure $\mathrm{Cov}(\mathcal{D}, \widehat{\pi}) \approx 81\%$[1], which is sufficient to maintain theoretical guarantees.

**Design Implications and Empirical Support**   Theorem 3.1 highlights three practical strategies for tightening the suboptimality bound: (1) improving annotation fidelity or scaling model capacity to reduce the value approximation error $\varepsilon$; (2) maintaining high dataset coverage by initializing the policy via supervised fine-tuning (SFT) on behavior-aligned trajectories, which mitigates distribution shift; and (3) freezing the learned value model $Q_\theta$ during policy optimization to stabilize gradients and enable tractable offline learning.

Empirically, we find that enhancing the quality of $Q_\theta$—via larger vision-language models or refined prompting—consistently improves downstream task success rates. For instance, scaling the value model from 7B to 32B parameters yields better action-value estimation and stronger policies. Additionally, policies trained without SFT exhibit degraded performance, underscoring the importance of warm-starting to remain within high-coverage regions of $\mathcal{D}$ and support the assumptions required for theoretical guarantees.

## 4 EXPERIMENTS

In this section, we evaluate the effectiveness of our proposed Value Environment Model (VEM) framework. We conduct experiments on two GUI automation benchmarks: AITW Rawles et al. (2024) representing mobile application environments, and MM-Mind2Web Deng et al. (2023) representing web-based environments.

---

[1]This data is calculated by Auto-GUI-Base in offline AITW benchmark

## 4.1 DATA COLLECTION

**Training Data for Critic Model.** To align Qwen2.5VL with the critic model's schema and evaluation framework, we used GPT-4o for multimodal data annotation and assessed agent actions. As shown in Figure 1, our evaluation system defines two action quality levels based on effectiveness.

**Level 1 (Suboptimal Actions).** Manifest deviations from optimal task execution, specifically including: (1) erroneous text inputs compromising workflow integrity, (2) interface interactions triggering adversarial outcomes such as advertisement redirections, and (3) premature declarations of task completion prior to objective fulfillment.

**Level 2 (Optimal Actions).** Demonstrate maximally effective task-solving behaviors, characterized by three critical patterns: (1) verifiable task completion through interface state validation, (2) robust recovery strategies, and (3) context-aware selection of optimal entry points for subtask resolution.

The complete evaluation prompts and annotation protocols are formally specified in Appendix J.

Table 1: AITW dataset: training/testing tasks and interaction steps with action quality levels.

| Category | Split | Tasks | Steps | L1 | L2 |
|---|---|---|---|---|---|
| General | Train | 436 | 3340 | 1187 | 2153 |
| | Test | 100 | 777 | 214 | 563 |
| Webshopping | Train | 560 | 6240 | 1939 | 4301 |
| | Test | 91 | 772 | 273 | 499 |

Table 2: MM-Mind2Web dataset composition: number of tasks and interaction steps.

| Split | Tasks | Steps |
|---|---|---|
| Train | 1009 | 7775 |
| Cross-Task Test | 252 | 2094 |
| Cross-Website Test | 177 | 1373 |
| Cross-Domain Test | 912 | 5911 |

Both benchmarks follow SeeClick Cheng et al. (2024) standards for data selection and evaluation. Tables 1 and 2 detail AITW and MM-Mind2Web task/step distributions and action quality levels (Level 1/2).

## 4.2 IMPLEMENTATION DETAILS

**Critic model** We used GPT-4o to annotate and score benchmark data with task descriptions, action sequences, evaluated actions, and annotated screenshots, enhancing labeling accuracy with visualized annotations. Suboptimal data was generated using GPT-4o to address data imbalances. Prompts are in Appendix J. More details are provided in Appendix E.

**Policy model** The AITW dataset policy is built on the SFT Auto-GUI base model, while the MM-Mind2Web policy uses Qwen2.5VL-3B as its base. In both cases, the critic model parameters are kept frozen during training. Our Q-value curve in training is very stable, as detailed in Appendix E.

## 4.3 VEM PERFORMANCE

We evaluate the performance of our VEM models on both the General and WebShopping datasets, as shown in Table 3. Our trained VEM achieves accuracy scores of 77% and 85% on the AITW and MM-Mind2Web benchmarks, demonstrating high performance reliability. This level of accuracy is sufficient to drive policy optimization via exploration. As shown in Figure 3, the model assigns a high value to the correct action ('click cart') while penalizing suboptimal actions, guiding the policy's exploration.

## 4.4 MAIN RESULTS

We evaluate our method on two benchmarks: AITW (General and Webshopping tasks) and MM-Mind2Web. For AITW, we use two evaluation schemes: (1) **Offline Evaluation** computes step/task success rates (SRs) by comparing predicted actions with human annotations; (2) **Online Evaluation** deploys the agent in Android environments (aligned with DigiRL), using GPT-4o as a judge, with 10-step limits and duplicate task removal. To ensure the reliability of this metric, we performed a manual validation study which revealed a 97.5% agreement rate between the LLM judge and human expert evaluations, supporting its use as an accurate proxy for task success (see Appendix H

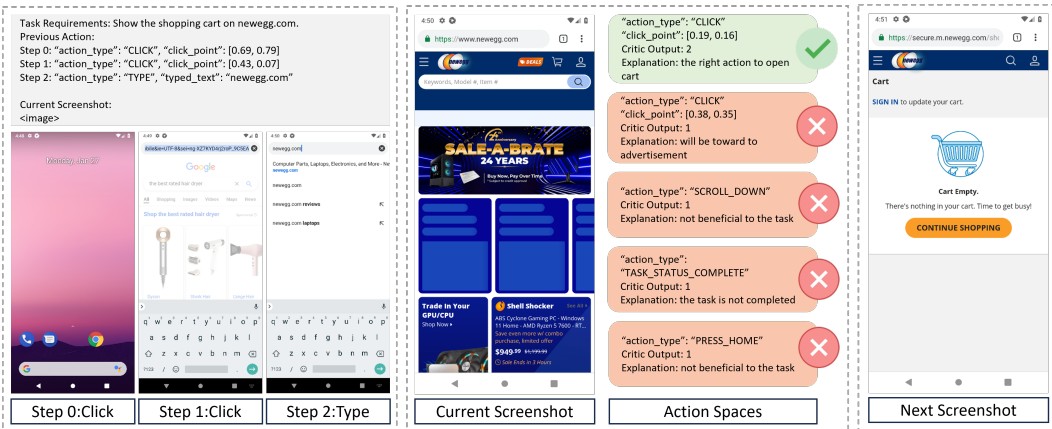

Figure 3: VEM scoring different actions at a single timestep.

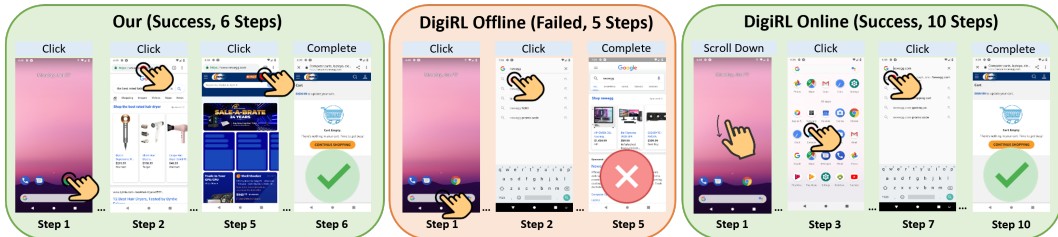

Figure 4: A case study of task execution trajectory comparison with DigiRL.

for details). For MM-Mind2Web, we conduct **Offline Evaluation**, calculating element accuracy, operation F1, and step success rate. Baselines Details can be found in Appendix E.

### 4.4.1 AITW BENCHMARK RESULTS

As shown in Table 4 and Table 5, our approach achieves SOTA performance on AITW benchmarks. Offline, our method with Auto-GUI policy model attains 30.0% Task SR in General and Webshopping do-

Table 3: Performance on the VEM.

| Dataset | Precision | Recall | F1 | Acc |
|---|---|---|---|---|
| AITW (General+Webshopping) | 0.83 | 0.84 | 0.83 | 0.77 |
| MM-Mind2Web | 0.85 | 0.84 | 0.84 | 0.85 |

mains, outperforming baselines. Scaled to SeeClick (9.6B), it reaches 34.0% and 35.0%, showing value model guidance efficacy without extra data. Online, our method generalizes well under real-world noise, achieving 49.15% and 20.00% success in General and Webshopping. Interaction efficiency is maintained with step lengths comparable to supervised methods (Auto-GUI: 7.83/7.86 vs. 7.92/9.34 , SeeClick: 7.59/7.68 vs. 8.80/9.80), avoiding DigiRL-online's trade-off of shorter steps (7.25/7.37) for lower success.

### 4.4.2 MIND2WEB BENCHMARK RESULTS

As shown in Table 6, our 3B model attains the highest Step Success Rate (Step SR) of 55.8%, 51.2%, and 50.5% under Cross-Task, Cross-Website, and Cross-Domain settings. Compared with other models trained only on the MM-Mind2Web dataset, our model achieves state-of-the-art (SOTA) performance across all metrics. Even when compared with models that have been extensively and diversely trained on larger datasets, our model shows only slight underperformance in operation F1 on two test sets, while still surpassing them in step success rate. This fully demonstrates the effectiveness of VEM, which can effectively simulate interaction with the environment on a limited training set, achieving excellent results even on a 3B policy model.

### 4.4.3 ABLATION STUDY

We performed extensive ablation studies (detailed in Appendix F) which confirmed the benefits of a larger critic model and more training data. Crucially, the studies showed that SFT initialization is

Table 4: Offline results on AITW benchmark.

|  | General | | Webshopping | |
| --- | --- | --- | --- | --- |
|  | Step SR | Task SR | Step SR | Task SR |
| GPT-4o OpenAI (2024) | 68.4 | 9.0 | 63.9 | 5.0 |
| Auto-GUI (200M) Zhang & Zhang (2024) | 83.3 | 20.0 | 78.0 | 18.0 |
| CogAgent (9B) Hong et al. (2024) | 73.7 | 16.0 | 72.2 | 9.0 |
| Seeclick (9.6B) Cheng et al. (2024) | 83.3 | 26.0 | 79.1 | 18.0 |
| Digirl-offline (200M) Bai et al. (2024) | 83.2 | 23.0 | 84.2 | 24.0 |
| Digirl-online (200M) Bai et al. (2024) | 83.3 | 26.0 | 85.1 | 26.0 |
| DigiQ (200M) Bai et al. (2025a) | 84.2 | 29.0 | 85.4 | 28.0 |
| Ours$_{\text{Auto-GUI}}$ (200M) | 84.7 | 30.0 | 85.6 | 30.0 |
| Ours$_{\text{SeeClick}}$ (9.6B) | **86.3** | **34.0** | **86.7** | **35.0** |

Table 5: Online results on AITW benchmark.

|  | General | | Webshopping | |
| --- | --- | --- | --- | --- |
|  | Task SR | Step Len | Task SR | Step Len |
| GPT-4o OpenAI (2024) | 0.00 | 9.00 | 0.00 | 9.91 |
| Auto-GUI (200M) Zhang & Zhang (2024) | 28.81 | 7.92 | 2.86 | 9.34 |
| CogAgent (9B) Hong et al. (2024) | 38.98 | 7.23 | 14.29 | 7.80 |
| Seeclick (9.6B) Cheng et al. (2024) | 25.42 | 8.80 | 11.43 | 9.80 |
| Digirl-offline (200M) Bai et al. (2024) | 38.98 | 7.61 | 14.29 | 8.26 |
| Digirl-online (200M) Bai et al. (2024) | 38.98 | 7.25 | 11.43 | 7.37 |
| DigiQ (200M) Bai et al. (2025a) | 33.90 | 6.76 | 5.71 | 8.43 |
| Ours$_{\text{Auto-GUI}}$ (200M) | 42.37 | 7.83 | 14.29 | 7.86 |
| Ours$_{\text{SeeClick}}$ (9.6B) | **49.15** | 7.59 | **20.00** | 7.68 |

Table 6: Performance comparison on MM-Mind2Web across different settings. We report element accuracy (Ele.Acc), operation F1 (Op.F1), and step success rate (Step SR).

| Method | Cross-Task | | | Cross-Website | | | Cross-Domain | | |
| --- | --- | --- | --- | --- | --- | --- | --- | --- | --- |
|  | Ele.Acc | Op.F1 | Step SR | Ele.Acc | Op.F1 | Step SR | Ele.Acc | Op.F1 | Step SR |
| **Agent Framework** | | | | | | | | | |
| GPT-4o + SeeClick Cheng et al. (2024)[†] | 32.1 | - | - | 33.1 | - | - | 33.5 | - | - |
| GPT-4o + UGround Gou et al. (2025)[†] | 47.7 | - | - | 46.0 | - | - | 46.6 | - | - |
| GPT-4V + SeeAct Zheng et al. (2024a) | 46.4 | 73.4 | 40.2 | 38.0 | 67.8 | 32.4 | 42.4 | 69.3 | 36.8 |
| GPT-4V + OmniParser Wan et al. (2024)[‡] | 42.4 | 87.6 | 39.4 | 41.0 | 84.8 | 36.5 | 45.5 | 85.7 | 42.0 |
| **Agent Model** | | | | | | | | | |
| GPT-4o OpenAI (2024)[‡] | 5.7 | 77.2 | 4.3 | 5.7 | 79.0 | 3.9 | 5.5 | 86.4 | 4.5 |
| GPT-4 (SOM) OpenAI et al. (2024)[‡] | 29.6 | - | 20.3 | 20.1 | - | 13.9 | 27.0 | - | 23.7 |
| MindAct-XL-3B Deng et al. (2023)[* §] | 55.1 | 75.7 | 52.0 | 42.0 | 65.2 | 38.9 | 42.1 | 66.5 | 39.6 |
| WebGUM-XL-3B Furuta et al. (2024)[* §] | 57.2 | 80.3 | 53.7 | 45.3 | 70.9 | 41.6 | 43.9 | 72.2 | 41.4 |
| SeeClick-9.6B Cheng et al. (2024)[§] | 28.3 | 87.0 | 25.5 | 21.4 | 80.6 | 16.4 | 23.2 | 84.8 | 20.8 |
| ShowUI-2B Lin et al. (2024) | 39.9 | **88.6** | 37.2 | 41.6 | 83.5 | 35.1 | 39.4 | 86.8 | 35.2 |
| Falcon-UI-7B Shen et al. (2024) | - | - | 31.7 | - | - | 25.8 | - | - | 25.2 |
| Magma-8B Yang et al. (2025) | 57.2 | 76.9 | 45.4 | 54.8 | 79.7 | 43.4 | 55.7 | 80.6 | 47.3 |
| MiniCPM-V-GUI-8B Chen et al. (2024) | 20.3 | 81.7 | 17.3 | 23.8 | **86.8** | 20.8 | 17.9 | 74.5 | 17.6 |
| Ours-3B | **61.5** | 85.7 | **55.8** | **58.9** | 86.5 | **51.2** | **58.2** | **87.4** | **50.5** |

[‡] These results come from Qin et al. (2025). [†] These results come from Gou et al. (2025). [§] These models, like Ours, were trained only on the training set of MM-Mind2Web, while other models were trained on more data. [*] Text-only input.

vital for performance, our binary labeling scheme was more effective than a finer-grained 3-class approach. It also includes an efficiency comparison against environment-based RL and analyses of different LLM annotators.

## 4.5 CASE STUDY

Real-world deployment of GUI agents faces challenges from dynamic environments, often causing navigation errors like entering ads. Models trained via SFT struggle to recover, while our value-guided approach enables robust adaptation. As shown in Figure 4, our method completes the task-"Show the shopping cart on newegg.com" via precise state valuation, avoiding common failures seen in DigiRL baselines. This is enabled by our value model's continuous feedback interpretation. Further comparisons are in Appendix K.

## 5 CONCLUSION

We presents an environment-free RL framework for GUI automation that decouples value estimation from policy optimization through a Value Environment Model (VEM). Our approach replaces error-prone next-state simulations with semantic reasoning over GUI elements, enabled by offline learning from human demonstration data. The two-stage training paradigm achieves structured credit assignment without environmental interaction, while maintaining procedural efficiency comparable to environment-based methods. Experimental results on Android-in-the-Wild and Multimodal-Mind2Web demonstrate superior task success rates over existing environment-free approaches and significant improvements in generalization capability compared to vision-language models. The framework establishes semantic-driven value estimation as an effective pathway for layout-agnostic GUI automation with sample efficiency. In the future, we plan to explore self-supervised approaches for training the value model, aiming to reduce labeling overhead and further improve scalability.

ETHICS STATEMENT

We acknowledge the ICLR Code of Ethics and confirm that our work adheres to its principles. Our research does not involve human subjects or personally identifiable information. The datasets we used are publicly available, and we strictly followed their licenses and usage policies. We have carefully considered potential risks of privacy leakage, bias, or unfairness. We believe our findings contribute positively to the research community without foreseeable harmful applications.

REPRODUCIBILITY STATEMENT

We are committed to ensuring the reproducibility of our work. All experimental settings, model architectures, and hyperparameters are detailed in Sections 3, 4 and E. Additional implementation details, data preprocessing procedures, and complete results are provided in the Appendix. We have released the source code in the anonymous link: https://anonymous.4open.science/r/VEM-Agent-51E7.

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

# APPENDIX

## A  THE USE OF LARGE LANGUAGE MODELS (LLMs)

In accordance with the ICLR policy on LLM usage, we hereby disclose that Large Language Models (LLMs) were only used as auxiliary tools for language polishing and minor grammatical improvements in the writing process. They were not involved in research ideation, experimental design, implementation, data analysis, or the generation of scientific content. The authors take full responsibility for the content of this paper.

## B  IMPACT STATEMENT

This work presents an environment-free framework for training GUI agents via a pretrained Value Environment Model (VEM), which offers several positive societal implications.

First, in the domain of GUI agents, the proposed method significantly lowers the barrier to developing and deploying intelligent agents across diverse user interfaces, including mobile and web environments. By eliminating the need for interactive environments during training, the framework enhances sample efficiency and robustness to layout variability, enabling broader adoption of automation technologies in areas such as accessibility support, software testing, and digital task assistance.

Second, from the perspective of reward modeling, VEM shifts the focus from environment-dependent rewards to semantically grounded value estimation. This approach provides stable and generalizable supervision signals, reducing reliance on brittle hand-crafted rewards or noisy environment feedback. It offers a scalable and interpretable alternative for aligning agent behavior with human intent in scenarios where explicit reward signals are unavailable or costly to obtain.

Third, with regard to world modeling, the VEM introduces a value-centric abstraction that bypasses the need for explicit state transition modeling. By learning long-term action utility directly from offline data, the method avoids compounding prediction errors common in traditional model-based approaches. This lightweight form of world modeling demonstrates strong generalization across tasks and environments, offering a promising direction for leveraging pretrained multimodal models as implicit world models.

In summary, this work contributes to the development of scalable, robust, and semantically aligned agents, advancing the broader goal of building efficient and general-purpose intelligent systems for real-world graphical user interfaces.

## C  ACTIONS

### C.1  AITW ACTIONS

The available actions include CLICK, TYPE, PRESS_BACK, PRESS_HOME, SCROLL_DOWN, SCROLL_UP, SCROLL_LEFT, SCROLL_RIGHT, PRESS_ENTER, STATUS_TASK_COMPLETE, and STATUS_TASK_IMPOSSIBLE.

Please note that the action space here is different from the origin action space in AITW. We have split the DUAL_POINT in AITW into two parts: click and scroll, specifically as follows: CLICK, SCROLL_DOWN, SCROLL_UP, SCROLL_LEFT, SCROLL_RIGHT.

### C.2  MM-MIND2WEB ACTIONS

The available actions include CLICK, TYPE, SELECT, this is consistant with the action space in MM-Mind2Web.

## D    PROOF OF EXTENDED PERFORMANCE BOUND

*Proof.* We bound the suboptimality gap $J(\pi^*) - J(\widehat{\pi})$ in two stages: (i) relating value-function approximation error to policy return difference, and (ii) accounting for distribution shift between $\widehat{\pi}$ and the behavior policy $\beta$.

**1. Relating Q-function Error to Return Difference.** By the performance-difference lemma Kakade & Langford (2002), for any two policies $\pi$ and $\pi'$,

$$J(\pi) - J(\pi') = \frac{1}{1-\gamma} \mathbb{E}_{s \sim d_\pi} \Big[ \mathbb{E}_{a \sim \pi(\cdot|s)} Q^{\pi'}(s,a) - V^{\pi'}(s) \Big].$$

Taking $\pi = \pi^*$ and $\pi' = \widehat{\pi}$, and noting $V^{\pi^*}(s) = \max_a Q^*(s,a)$, we get

$$J(\pi^*) - J(\widehat{\pi}) = \frac{1}{1-\gamma} \mathbb{E}_{s \sim d_{\pi^*}} \Big[ \max_a Q^*(s,a) - \mathbb{E}_{a \sim \widehat{\pi}} Q^*(s,a) \Big].$$

Under our approximate-Q assumption,

$$\big| Q_\theta(s,a) - Q^*(s,a) \big| \leq \varepsilon \quad \forall (s,a) \in \mathcal{D}.$$

Since $\widehat{\pi}$ maximizes $\mathbb{E}_{s \sim \mathcal{D}, a \sim \pi}[Q_\theta(s,a)]$, for each $s \in \mathcal{D}$,

$$\max_a Q_\theta(s,a) - \mathbb{E}_{a \sim \widehat{\pi}} Q_\theta(s,a) \ \leq \ 0.$$

Hence for all $s \in \mathcal{D}$,

$$\max_a Q^*(s,a) - \mathbb{E}_{a \sim \widehat{\pi}} Q^*(s,a)$$

$$\leq \big[ \max_a Q_\theta(s,a) - \mathbb{E}_{a \sim \widehat{\pi}} Q_\theta(s,a) \big] + 2\varepsilon \ \leq \ 2\varepsilon.$$

Thus, restricting the expectation in the performance-difference lemma to $s \in \mathcal{D}$,

$$J(\pi^*) - J(\widehat{\pi}) \ \leq \ \frac{2\varepsilon}{1-\gamma} \, \mathbb{P}(s \in \mathcal{D} \mid s \sim d_{\pi^*}).$$

Under coverage condition, $\widehat{\pi}$ does not visit unseen $(s,a)$, so $\mathcal{D}$ covers the support of $d_{\pi^*}$, and the above probability is (approximately) 1.

**2. Accounting for Distribution Shift.** In practice, $\widehat{\pi}$ may induce a state distribution $d_{\widehat{\pi}}$ differing from $d_{\pi^*}$. Standard distribution-shift bounds (see, e.g., Levine et al. (2020)) yield

$$\big| J(\pi^*) - J(\widehat{\pi}) \big| \ \leq \ \frac{2\varepsilon}{1-\gamma} + \frac{R_{\max}}{1-\gamma} \, \| \, d_{\pi^*} - d_{\widehat{\pi}} \|,$$

where $R_{\max} = \max_{s,a} |r(s,a)|$ and $\| \cdot \|$ is a divergence measure. Since $r(s,a)$ is bounded by $Q^*(s,a) \leq Q_{\max}$, we absorb constants into $c$. Moreover, $\|d_{\pi^*} - d_{\widehat{\pi}}\|$ can be bounded by $\|\widehat{\pi} - \beta\|$ under mixing conditions (cf. Petrik & Scherrer (2008)).

**Conclusion.**    Combining the two steps, there exists a constant $c > 0$ (depending on $\gamma$, $Q_{\max}$, and mixing properties) such that

$$J(\pi^*) - J(\widehat{\pi}) \ \leq \ c \left( \varepsilon + \|\widehat{\pi} - \beta\| \right).$$

Finally, because $\widehat{\pi}$ is trained purely offline with a fixed $Q_\theta$, its gradient estimates rely on deterministic VEM queries rather than noisy environment samples, yielding lower variance compared to on-policy RL.

This completes the proof.    □

## E    IMPLEMENTATION DETAILS

**Data Format**    The reinforcement learning paradigm requires standardized data transformations across input modalities. For our experiments on the AITW dataset, we trained the Auto-GUI-Base model, which is pre-trained to output normalized coordinates in the range of [0,1]. Therefore, we utilize a device-agnostic [0,1] screen-space coordinate system. For the MM-Mind2Web dataset, we trained and experimented with the Qwen2.5-VL-3B model, which tends to produce absolute coordinates rather than normalized [0,1] coordinates, so we used the original coordinate space to achieve better performance. Detailed action space configurations are provided in Appendix C.

**Critic model** We employ GPT-4o to annotate and score data within the benchmark training set. The input provided to GPT-4o comprised the task description, the complete sequence of actions, the specific action that requires evaluation, and the corresponding annotated screenshot. The inclusion of the global action sequence, along with visualized annotations on the screenshot (indicating click coordinates), enhanced the accuracy of GPT-4o's automated labeling. Furthermore, acknowledging inherent data quality issues within the benchmark dataset, which includes a small proportion of sub-optimal steps, we proactively generated additional suboptimal data using GPT-4o. This augmentation strategy aimed to mitigate potential biases in the Critic Model's training due to skewed data score distributions. The prompts utilized for data annotation and suboptimal sample generation by GPT-4o are detailed in Appendix J.

To evaluate GPT-4o's annotation quality on the benchmarks, we randomly sampled 50 instances from each dataset and had them independently annotated by three human experts. Using majority voting as the human ground truth, we found GPT-4o annotations to match with 90% accuracy. GPT-4o annotations on both benchmarks achieve 90% human consistency with 3-hour processing efficiency. Evaluation details can be found in I.

We fine-tuned the Qwen2.5VL-7B model using the LLaMA-Factory Zheng et al. (2024b) framework to develop state classification capabilities. The input part of our Critic Model includes: the textual task description, the history of actions, the current screen image, and the action currently pending execution. The output of our Critic Model is the score given by GPT-4o. The formal specification of input composition and prompting strategy appears in Appendix J.

Training leveraged distributed data parallelism on an 4-GPU NVIDIA A100 cluster, configured with 3 training epochs and a global batch size of 16. The optimization process employed AdamW with an initial learning rate of $1 \times 10^{-5}$, achieving convergence within 12 hours while maintaining computational efficiency through gradient accumulation strategies.

**Policy model** In our experiments, we found that training with reinforcement learning without prior supervised fine-tuning leads to poor performance. We attribute this to two main reasons: (1) prompts alone are insufficient to ensure correct output formatting, and (2) the model's initial capability is weak, resulting in extremely sparse positive samples during training, which makes effective learning difficult. In training the critic model, we enhance the quality of annotated data by providing GPT-4o with the global action sequence and the annotated current screenshot. Additionally, we construct suboptimal negative samples using GPT-4o to mitigate the issue of label bias. These strategies collectively contribute to improving the quality of $Q_\theta$.

For the AITW dataset, the policy architecture builds upon the Supervised Fine-Tuned (SFT) Auto-GUI base model. For the MM-Mind2Web dataset, we used Qwen2.5VL-3B as the base model. During the training of our policy model, we kept the parameters of the critic model frozen.

For the MM-Mind2Web dataset, we first used Qwen2.5VL-3B as the base model and performed supervised fine-tuning using the LLaMA-Factory Zheng et al. (2024b) framework with a learning rate of 1e-5 for one epoch, enabling the model to learn the output format. This was followed by reinforcement learning training.

Our implementation executed full-parameter optimization on an 4-GPU NVIDIA A100 cluster, configuring the training process with a batch size of 16 samples and $1 \times 10^{-5}$ across 10-20 training epochs. This configuration achieved stable convergence through progressive reward signal alignment, demonstrating parameter-efficient adaptation characteristics.

Figure 5 shows the Q-value curves during training of the policy model based on Auto-GUI for the AITW dataset and the policy model based on Qwen2.5VL-3B for the MM-Mind2Web dataset. As observed, the Q-values initially increase and then begin to converge. From the figure, it's evident that our training is very stable.

Please note that the Q-value discussed herein has undergone a normalization procedure. Specifically, the Critic Model yields Q-values that are either 1 or 2. Subsequently, a scaling transformation is applied according to the formula:

$$Q_{scaled} = \frac{Q}{2} - 0.75$$

This transformation effectively maps the original Q-values into a bounded range of $[-0.25, 0.25]$.

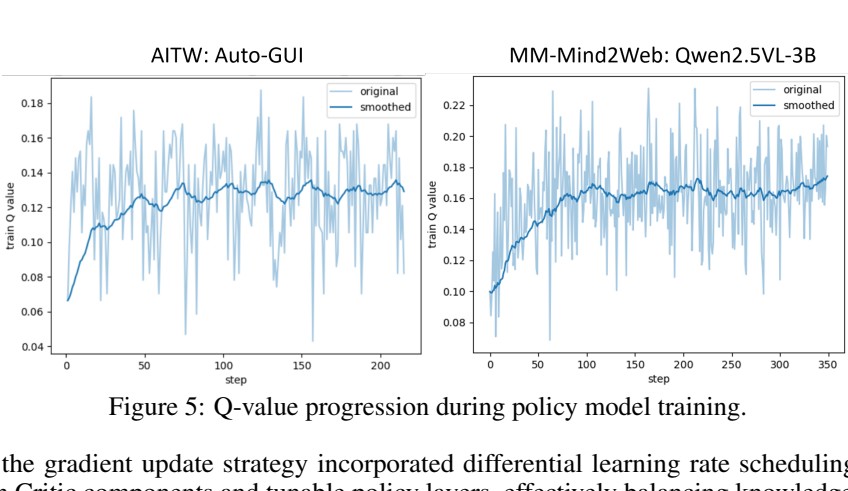

Figure 5: Q-value progression during policy model training.

Notably, the gradient update strategy incorporated differential learning rate scheduling between the frozen Critic components and tunable policy layers, effectively balancing knowledge retention with operational flexibility. The training of the Auto-GUI model required under 12 hours, while the training of the Qwen2.5VL-3B model required under 36 hours, reflecting optimized memory utilization patterns.

**Datasets** The Android in the Wild (AITW) dataset is a large-scale benchmark designed for research in Android device control via natural language instructions. It comprises 715k episodes across 30k unique instructions, collected from over 350 Android applications and websites. The dataset includes interactions recorded on eight device types (from Pixel 2 XL to Pixel 6) running Android versions 10 through 13, encompassing various screen resolutions.

Following the setup adopted in SeeClick Cheng et al. (2024), we selected a subset of data from the General and WebShopping categories of the AITW dataset to serve as our training and testing sets. This selection was made to align with the instruction-wise split strategy proposed by SeeClick, which aims to mitigate overfitting and better assess generalization capabilities across diverse tasks and applications.

The MM-Mind2Web benchmark comprises over 2k open-ended tasks collected from 137 real-world websites across 31 domains. MM-Mind2Web features a training set and three test sets: (1)Cross Task, which includes tasks from websites seen during training; (2)Cross Website, which contains tasks from unseen websites; (3)Cross Domain, which comprises tasks from entirely unseen domains—designed to evaluate different aspects of model generalization.

**Baselines** For AITW benchmark, to comprehensively evaluate our proposed method, we compare it against several baselines on the Android-in-the-Wild (AITW) benchmark. These baselines encompass a diverse range of approaches in multimodal reasoning, visual-language modeling, and reinforcement learning for GUI-based agents.

- **GPT-4o** OpenAI (2024) is OpenAI's flagship multimodal model that integrates text, vision, and audio modalities. Despite its general-purpose capabilities, GPT-4o exhibits limited performance on GUI navigation tasks, highlighting the challenges of applying generalist models to specialized domains.
- **Auto-GUI** Zhang & Zhang (2024) introduces a multimodal chain-of-action framework that leverages visual context and historical action sequences to predict subsequent actions. Trained on the AITW dataset, Auto-GUI achieves competitive performance in both offline and online settings.
- **CogAgent** Hong et al. (2024) is a visual-language model designed for GUI understanding and navigation. By employing high-resolution image encoders, CogAgent effectively captures fine-grained UI elements, leading to improved performance on benchmarks like AITW and Mind2Web.
- **SeeClick** Cheng et al. (2024) focuses on GUI grounding by pretraining on the ScreenSpot dataset, which includes diverse screenshots and instructions. As a purely vision-based agent, SeeClick demonstrates strong performance on GUI tasks with minimal training data, emphasizing the efficacy of GUI grounding pretraining.

- **DigiRL** Bai et al. (2024) employs an offline reinforcement learning approach to train device-control agents. By utilizing advantage-weighted RL and an automatic curriculum, DigiRL significantly improves success rates on the AITW dataset compared to supervised fine-tuning methods.

- **DigiQ** Bai et al. (2025a) builds upon DigiRL by introducing Q-value function learning for visual-language models. This approach enhances data efficiency and convergence performance, enabling better policy extraction and improved success rates on GUI navigation tasks.

For MM-Mind2Web benchmark, we evaluate our approach against a diverse set of baselines, encompassing both agent frameworks and agent models. These baselines are selected to represent the current state-of-the-art in multimodal web agents, covering a range of modalities and grounding strategies.

**Agent Frameworks**

- **GPT-4o + SeeClick** Cheng et al. (2024): This framework utilizes GPT-4o for planning and SeeClick for visual grounding. SeeClick is a vision-based grounding method that identifies clickable elements on the UI using object detection techniques.

- **GPT-4o + UGround** Gou et al. (2025): UGround is a universal grounding framework that combines textual and visual cues to map instructions to UI elements. When paired with GPT-4o, it enhances the agent's ability to interpret and act upon complex web interfaces.

- **GPT-4V + SeeAct** Zheng et al. (2024a): SeeAct integrates GPT-4V with a grounding strategy that leverages both HTML structure and visual information. It demonstrates strong performance in executing tasks on live websites by effectively grounding textual plans into actionable steps.

- **GPT-4V + OmniParser** Wan et al. (2024): OmniParser is a vision-based UI parser that converts screenshots into structured representations. Combined with GPT-4V, it enables the agent to understand and interact with web interfaces using visual inputs alone.

**Agent Models**

- **GPT-4o** OpenAI (2024): GPT-4o is a cutting-edge multimodal large language model engineered to process and generate responses across text, vision, and audio modalities. Its robust architecture enables it to serve as a powerful baseline for evaluating sophisticated multimodal understanding and the generation of contextually relevant actions in diverse scenarios.

- **GPT-4 (SOM)** OpenAI et al. (2024): This approach leverages the capabilities of the GPT-4 large language model, enhanced by the Set-of-Marks (SOM) prompting technique. SOM aims to significantly improve the model's grounding abilities, particularly in tasks that require precise identification and interaction with specific user interface (UI) elements within visual inputs.

- **MindAct-XL-3B** Deng et al. (2023): A 3B-parameter model for web automation, excelling at generating executable actions (clicks, typing, navigation) from webpage visuals and natural language. Trained on MM-Mind2Web, it shows strong performance on the Mind2Web benchmark.

- **WebGUM-XL-3B** Furuta et al. (2024): A 3B-parameter multimodal agent processing webpage screenshots and HTML for web navigation (clicking, typing). Jointly fine-tuned on instruction-finetuned LM and vision encoder, achieving SOTA on MiniWoB and WebShop, with strong generalization to Mind2Web.

- **SeeClick-9.6B** Cheng et al. (2024): A visual GUI agent automating tasks using only screenshots, focusing on GUI grounding. It employs GUI grounding pre-training and automated data curation, introducing ScreenSpot, a GUI grounding benchmark. Results show a correlation between GUI grounding and downstream task performance.

- **ShowUI-2B** Lin et al. (2024): A 2B-parameter vision-language-action model for GUI automation. Key features include UI-guided visual token selection and interleaved vision-language-action streaming. Trained on a 256K instruction-following dataset.

- **Falcon-UI-7B** Shen et al. (2024): A GUI agent model enhancing GUI context understanding. It introduces Insight-UI Dataset for instruction-free pre-training of GUI comprehension, followed by fine-tuning on instruction-based datasets.

- **Magma-8B** Yang et al. (2025): A foundation model for multimodal AI agentic tasks, integrating verbal and spatial-temporal intelligence. Pre-trained on diverse datasets with Set-of-Mark (SoM) for action grounding and Trace-of-Mark (ToM) for action planning, excelling in UI navigation and robotic manipulation.

- **MiniCPM-V-GUI-8B** Chen et al. (2024): A visual-based GUI agent model trained on the GUICourse dataset suite. Based on MiniCPM-V, it enhances VLM's interaction with GUIs, improving performance in OCR, grounding, and understanding GUI components and interactions.

# F    ABLATION STUDY

To comprehensively evaluate our VEM framework and validate its key design choices, we conducted an extensive series of ablation studies and analyses. These experiments investigate the impact of data scale, model size, the supervision signal itself, and the overall algorithmic contribution and efficiency of our approach.

## F.1    IMPACT OF CORE COMPONENTS AND TRAINING STRATEGY

We first analyze the impact of fundamental components: the scale of training data, the size of the critic and policy models, and the necessity of Supervised Fine-Tuning (SFT) initialization.

**Data and Model Scaling.** As shown in Table 7, performance consistently improves with more VEM training data and larger critic/policy models. For instance, increasing the critic model from 7B to 32B parameters boosts the General Task SR. Notably, even with only 30% of the training data, our method remains effective. This demonstrates the framework's robustness and scalability.

**Importance of SFT Initialization.** To quantify the role of SFT, we trained a policy using only VEM-guided RL without the SFT warm-start. Table 8 shows a substantial drop in performance on the MM-Mind2Web benchmark. This confirms that SFT is a crucial step for guiding the policy into a reasonable region of the state-action space and mitigating distribution shift, a common practice in the field.

Table 7: Ablation study on data and model scaling (AITW offline benchmark). The asterisk (*) marks the baseline configuration for each study.

| Ablation Var | Configuration | General | | Webshopping | |
|---|---|---|---|---|---|
| | | Step SR (%) | Task SR (%) | Step SR (%) | Task SR (%) |
| VEM Training Data | Full Dataset (100%)* | 84.7 | 30.0 | 85.6 | 30.0 |
| | Reduced (50%) | 84.1 | 28.0 | 85.2 | 28.0 |
| | Minimal (30%) | 83.4 | 26.0 | 84.0 | 27.0 |
| Critic Model Size | Qwen2.5VL-7B* | 84.7 | 30.0 | 85.6 | 30.0 |
| | Qwen2.5VL-32B | 85.7 | 32.0 | 86.2 | 32.0 |
| Policy Model Size | Ours (200M)* | 84.7 | 30.0 | 85.6 | 30.0 |
| | Ours (9.6B) | 86.3 | 34.0 | 86.7 | 35.0 |

Table 8: Ablation on SFT initialization (MM-Mind2Web benchmark). Performance degrades significantly without the SFT warm-start.

| Method | Cross-Task(%) | Cross-Website(%) | Cross-Domain(%) |
|---|---|---|---|
| SFT + VEM RL (Ours) | **55.8** | **51.2** | **50.5** |
| VEM RL only (no SFT) | 37.5 | 41.7 | 39.1 |

## F.2 ANALYSIS OF THE VEM SUPERVISION SIGNAL

The quality of the VEM is contingent on the supervision signal used to train it. We analyzed several factors, including the granularity of the value labels and the methodology for generating them.

**Value Label Granularity.** We experimented with a finer-grained 3-class labeling scheme (Low, Medium, High). As shown in Table 9, the simpler binary labeling approach outperformed the 3-class scheme on the AITW benchmark. This suggests that the binary distinction provides a more stable and effective supervision signal, likely because most actions in the dataset are either clearly beneficial or not, making the intermediate category sparse and difficult to learn from.

**Annotator and Prompt Design.** The quality of LLM-based annotation is critical. We evaluated the impact of the LLM annotator and the prompt design on annotation accuracy for MM-Mind2Web, using the dataset's ground truth as a reference. As detailed in Table 10, using a more powerful LLM (GPT-4o) and providing the full trajectory context in the prompt both lead to higher-quality supervision. This validates our chosen methodology for generating reliable value labels.

Table 9: Comparison of binary vs. 3-class value labels on the AITW offline benchmark.

| Label Type | Step SR (%) | Task SR (%) |
|---|---|---|
| Binary (Ours) | **83.6** | **29.0** |
| 3-Class | 82.9 | 25.0 |

Table 10: Analysis of annotation accuracy on MM-Mind2Web based on LLM annotator and prompt design.

| Configuration | Annotation Accuracy (%) |
|---|---|
| **Annotator Choice** | |
| GPT-4o (Ours) | **96.39** |
| Qwen2.5VL-32B | 93.45 |
| **Prompt Design** | |
| Full Trajectory (Ours) | **96.39** |
| Step-only Context | 62.57 |

## F.3 ALGORITHMIC CONTRIBUTION AND EFFICIENCY

Finally, we conducted experiments to isolate the contribution of our algorithm and quantify its efficiency gains.

**Isolating Algorithmic Contribution.** To verify that our performance gains stem from the VEM framework itself, not just the base model, we compared our method against Digi-Q using an identical LLaVA-1.5-7B backbone. As shown in Table 11, even on a level playing field, our method significantly outperforms Digi-Q, confirming the algorithmic advantage of our value-estimation approach.

**Efficiency Analysis.** To quantify the practical benefits of our environment-free approach, we compared its computational and sample costs against a representative environment-based method, DigiRL (Table 12). Our VEM framework achieves superior or comparable performance while being significantly more efficient, entirely avoiding the time, cost, and complexity of live environment rollouts.

## G LIMITATIONS

Our approach requires training a Value Environment Model, which in turn necessitates additional high-quality data annotation. In this work, we leverage GPT-4o to perform the annotation task. By providing GPT-4o with a detailed task description, the complete action history, the current action,

Table 11: Comparison with Digi-Q on the AITW online benchmark using an identical LLaVA-1.5-7B backbone.

| Method | General | | Webshopping | |
|---|---|---|---|---|
| | Task SR (%) | Step Len | Task SR (%) | Step Len |
| Digi-Q (LLaVA-7B) | 33.90 | 6.76 | 5.71 | 8.43 |
| Ours (LLaVA-7B) | **40.68** | 7.91 | **14.29** | 8.12 |

Table 12: Efficiency comparison between our offline VEM framework and an environment-based RL method.

| Metric | VEM (Ours) | DigiRL (Env-based) |
|---|---|---|
| Training Time | ~12 hours (offline) | ~3-5 days (with env rollouts) |
| API Calls for Supervision | ~15k (for annotation) | N/A (uses env rewards) |
| Total Samples Used | ~58k (offline) | >150k (online + offline) |

and the annotated screenshot corresponding to the current action, we are able to generate high-quality annotations that achieve up to 90% agreement with human experts.

Moreover, due to the limited number of negative samples in the dataset, there is a risk of label bias affecting the training of the Critic Model. To mitigate this issue, we also utilize GPT-4o to construct high-quality negative samples. These designs significantly improve the annotation accuracy of the Critic Model, albeit at the cost of requiring additional annotation resources. Exploring how to achieve comparable performance with fewer additional resources, or even in an unsupervised manner, remains an important direction for future work.

## H  VALIDATION OF LLM-BASED ONLINE EVALUATION

While oracle-based evaluation functions represent the gold standard for benchmarks, our online evaluation for AITW relies on an LLM-based judge (GPT-4o). This approach was chosen specifically for fair and direct comparability with recent state-of-the-art methods such as DigiRL and DigiQ, which are evaluated in the same manner.

To directly address and verify the reliability of this LLM-based evaluation, we conducted an additional manual validation study. We randomly sampled 40 task trajectories from our online experiments and had expert human annotators independently review the final outcomes determined by the LLM judge.

The results, summarized in Table 13, show a 97.5% agreement rate between the LLM's judgments and human experts. The LLM was highly accurate, particularly in identifying failures. This high degree of agreement strongly supports the reliability of using GPT-4o as a practical and accurate proxy for task success in the AITW-online setting.

Table 13: Human Validation of the LLM Judge for AITW-online Evaluation.

| Task Outcome (LLM Judge) | Number of Samples | Human Expert Confirmation |
|---|---|---|
| Failure | 29 | 29 Confirmed Failures |
| Success | 11 | 10 Confirmed Successes |
| Total | 40 | 39 Agreements (97.5%) |

## I  RESEARCH ON THE AUTOMATED ANNOTATION QUALITY OF GPT-4O

We conducted a systematic study on the annotation quality of GPT-4o, aiming to evaluate its consistency and accuracy on standard benchmark tasks. Specifically, we randomly sampled 50 instances from each of two public datasets—AITW and MM-Mind2Web. Three professional annotators, who

are proficient in both Android and Web platforms and have experience with GUI Agents, were invited to independently annotate each instance. During the annotation process, all annotators completed their work independently without knowledge of each other's labels. The annotation platform provided a unified interface and a detailed instruction document, which included example data, annotation criteria, and strategies for resolving common ambiguities.

We adopted a majority voting strategy to generate the human reference labels (the "gold standard") and compared them against the automated annotations produced by GPT-4o. The results show that GPT-4o achieved 90% agreement with the human annotations across both tasks. Moreover, it completed all annotations in approximately 3 hours, demonstrating strong efficiency and practical potential.

Regarding ethics, this study was approved by the Institutional Review Board (IRB) of our institution. There were no foreseeable physiological or psychological risks to participants, and all annotators provided informed consent before participation, fully understanding the nature and objectives of the study. Annotators were compensated on an hourly basis at a rate not lower than the local minimum wage, and in accordance with the average pay standards for professional data annotators. This complies with the NeurIPS Code of Ethics regarding fair compensation for labor involved in data collection and curation.

## J  PROMPT

**Prompt of AITW GPT-4o input**

```
As an expert in the field of GUI and reinforcement learning,
    you will receive complete screenshots and textual
    descriptions of interactions for a given task. You need
    to evaluate a specific step in terms of its value within
    the task chain, similar to what a value function does in
    reinforcement learning. Detailed criteria and standards
    are given below.

## Explanation of the input content:
1. Task: Brief description of the current GUI task, such as
    implementing the "Get Hong Kong hotel prices" task in
    Android GUI.
2. Complete operation description and corresponding
    screenshot sequence for the task
    (1) Text description of operations: Contains 11 types of
        GUI operations. Specific fields and their meanings are
         as follows:
    [1] CLICK: Click on a specific position on the screen.
        If it is a link or software, it will enter; if it
        is text, it will be selected. The "click_point" is
        represented by a two-dimensional array indicating
        the position of the click, relative to the top-left
         corner of the screenshot and within a range from
        0.0 to 1.0.
       - example: "action_type": "CLICK", "click_point":
          [0.5, 0.5]
    [2] TYPE: An action type that sends text. Note that
        this simply sends text and does not perform any
        clicks for element focus or enter presses for
        submitting text.
       - example: "action_type": "TYPE", "typed_text": "
          capital of England"
```

```
        [3] PRESS_BACK: Return to the previous page. Usually
            the previous webpage.
          - example: "action_type": "PRESS_BACK"
        [4] PRESS_HOME: Return to the system home page. Use
            this action to return to the home screen when the
            current screen is not the desired one, so you can
            reselect the program you need to enter.
          - example: "action_type": "PRESS_HOME"
        [5] PRESS_ENTER: Press the enter key to execute a step.
            Generally, after confirming the input text, use
            this action to start the search.
          - example: "action_type": "PRESS_ENTER"
        [6] STATUS_TASK_COMPLETE: An action used to indicate
            that the desired task has been completed and resets
            the environment. This action should also be used
            if the task is already completed and there is
            nothing more to do. For example, the task is to
            turn on the Wi-Fi when it is already on.
          - example: "action_type": "STATUS_TASK_COMPLETE"
        [7] STATUS_TASK_IMPOSSIBLE: An action used to indicate
            that the desired task is impossible to complete and
            resets the environment. This can result from
            various reasons including UI changes, Android
            version differences, etc.
          - example: "action_type": "STATUS_TASK_IMPOSSIBLE"
        [8] SCROLL_DOWN: Scroll down.
          - example: "action_type": "SCROLL_DOWN"
        [9] SCROLL_UP: Scroll up.
          - example: "action_type": "SCROLL_UP"
        [10] SCROLL_LEFT: Scroll left.
          - example: "action_type": "SCROLL_LEFT"
        [11] SCROLL_RIGHT: Scroll right.
          - example: "action_type": "SCROLL_RIGHT"
    (2) Corresponding screenshot before each operation. If the
         operation is of the "CLICK" type, the click position
         is marked with a red dot in the image.
  3. The current action to be evaluated and the corresponding
     screenshot.

  ## Evaluation Criteria:
  Here are the detailed descriptions of the two levels.
     Attention needs to be paid to whether the action taken
     based on the current screenshot promotes efficient task
     execution, rather than the relevance of the content shown
      in the current screenshot to the task:
  Level 1: The action is not the optimal choice for
     completing the task at this moment, which may lead to
     deviations from the task flow. For example:
     (1) Incorrect text input.
     (2) Clicking a button that might lead to an
         advertisement.
     (3) Announcing the task's success when it has not
         actually been achieved.
  Level 2: The action is the optimal and correct choice for
     completing the task at this moment. For example:
     (1) When showing task completion, the displayed content
          can fully achieve it.
```

```
        (2) When entering an unrelated interface, you can
            return to the main screen by executing "PRESS_HOME
            ."
        (3) Selecting the most correct entry point to complete
            the current task.

## Output requirements:
- Format: {"rating": int, "explanation": str}. Do not include
    any additional characters beyond this format
- The "rating" field should be represented by the number 1 or
     2 indicating the evaluation level. The "explanation"
    field should explain the evaluation process that led to
    this rating, without including descriptions of operations
     after the current step (future operations are considered
     unknown).

## Example Input:
Task Requirements: What is the capital of England?
Action and ScreenShot:
step 0: "action_type": "CLICK", "click_point": "[0.524,
    0.06]"
step 1: "action_type": "TYPE", "typed_text": "capital of
    England"
step 2: "action_type": "PRESS_ENTER"
step 3: "action_type": "STATUS_TASK_COMPLETE"
Current Action:
step 2: "action_type": "PRESS_ENTER"

## Example Output:
{"rating": 2, "explanation": "The action of pressing enter
    after typing 'capital of England' is an appropriate step
    to get the answer to the task requirement of finding out
    the capital of England, which is an optimal action
    towards achieving the task goal."}

Task Requirements: {}
Action and ScreenShot: {}
Current Action:
{}
```

**Prompt of AITW critic input**

```
As an expert in the field of GUI and reinforcement learning,
    you will receive textual descriptions of history
    interactions for a given task. You need to evaluate the
    current action, similar to what a value function does in
    reinforcement learning. Detailed criteria and standards
    are given below.

## Explanation of the input content:
1. Task: Brief description of the current GUI task, such as
    implementing the "Get Hong Kong hotel prices" task in
    Android GUI.
2. Description of History operation
```

```
    Contains 11 types of GUI operations. Specific fields and
        their meanings are as follows:
    [1] CLICK: Click on a specific position on the screen. If
        it is a link or software, it will enter; if it is text
        , it will be selected. The "click_point" is
        represented by a two-dimensional array indicating the
        position of the click, relative to the top-left corner
         of the screenshot and within a range from 0.0 to 1.0.
        - example: "action_type": "CLICK", "click_point": [0.5,
            0.5]
    [2] TYPE: An action type that sends text. Note that this
        simply sends text and does not perform any clicks for
        element focus or enter presses for submitting text.
        - example: "action_type": "TYPE", "typed_text": "
          capital of England"
    [3] PRESS_BACK: Return to the previous page. Usually the
        previous webpage.
        - example: "action_type": "PRESS_BACK"
    [4] PRESS_HOME: Return to the system home page. Use this
        action to return to the home screen when the current
        screen is not the desired one, so you can reselect the
         program you need to enter.
        - example: "action_type": "PRESS_HOME"
    [5] PRESS_ENTER: Press the enter key to execute a step.
        Generally, after confirming the input text, use this
        action to start the search.
        - example: "action_type": "PRESS_ENTER"
    [6] STATUS_TASK_COMPLETE: An action used to indicate that
        the desired task has been completed and resets the
        environment. This action should also be used if the
        task is already completed and there is nothing more to
         do. For example, the task is to turn on the Wi-Fi
        when it is already on.
        - example: "action_type": "STATUS_TASK_COMPLETE"
    [7] STATUS_TASK_IMPOSSIBLE: An action used to indicate
        that the desired task is impossible to complete and
        resets the environment. This can result from various
        reasons including UI changes, Android version
        differences, etc.
        - example: "action_type": "STATUS_TASK_IMPOSSIBLE"
    [8] SCROLL_DOWN: Scroll down.
        - example: "action_type": "SCROLL_DOWN"
    [9] SCROLL_UP: Scroll up.
        - example: "action_type": "SCROLL_UP"
    [10] SCROLL_LEFT: Scroll left.
        - example: "action_type": "SCROLL_LEFT"
    [11] SCROLL_RIGHT: Scroll right.
        - example: "action_type": "SCROLL_RIGHT"
3. The current action to be evaluated and the corresponding
   screenshot(the screenshot before each operation. If the
   operation is of the "CLICK" type, the click position is
   marked with a red dot in the image.)

## Evaluation Criteria:
Here are the detailed descriptions of the two levels.
    Attention needs to be paid to whether the action taken
    based on the current screenshot promotes efficient task
```

```
    execution, rather than the relevance of the content shown
      in the current screenshot to the task:
    Level 1: The action is not the optimal choice for
        completing the task at this moment, which may lead to
        deviations from the task flow. For example:
        (1) Incorrect text input.
        (2) Clicking a button that might lead to an
            advertisement.
        (3) Announcing the task's success when it has not
            actually been achieved.
    Level 2: The action is the optimal and correct choice for
        completing the task at this moment. For example:
        (1) When showing task completion, the displayed content
             can fully achieve it.
        (2) When entering an unrelated interface, you can
            return to the main screen by executing "PRESS_HOME
            ."
        (3) Selecting the most correct entry point to complete
            the current task.

## Output requirements: 1 or 2 (INT)

## Example Input:
Task Requirements: What is the capital of England?
Previous Action:
step 0: "action_type": "CLICK", "click_point": "[0.524,
    0.06]"
step 1: "action_type": "TYPE", "typed_text": "capital of
    England"
Current Action and Screenshot:
step 2: "action_type": "PRESS_ENTER"

## Example Output:
2

Task Requirements: {}
Previous Action:
{}
Current Action and Screenshot:
<image>
{}
```

**Prompt of MM-Mind2Web GPT-4o input**

```
As an expert in web interaction and reinforcement learning,
    you will receive a complete sequence of web interaction
    steps and corresponding descriptions for a given task.
    You need to evaluate a specific step in terms of its
    value within the task chain, similar to a value function
    in reinforcement learning. Detailed criteria and
    standards are given below.

## Explanation of the input content:
1. Task: Brief description of the current web task, such as "
    Search for a product on an e-commerce website".
```

```
2. Complete operation description and corresponding sequence
   for the task:
   (1) Text description of operations: Contains 3 types of
       web actions. Specific fields and their meanings are as
        follows:
       [1] CLICK: Click on a web element at a specific
           position. The "click_point" is represented by a two
           -dimensional array indicating the absolute position
            of the click in pixels.
         - example: "action_type": "click", "click_point":
            [100, 150]
       [2] TYPE: Click and input text into a field at a
           specific position. The "click_point" is represented
            by a two-dimensional array indicating the absolute
            position of the click in pixels.
         - example: "action_type": "type", "click_point":
            [200, 300], "value": "search term"
       [3] SELECT: Click at a specific position to open a
           dropdown menu, then select an option. Note: The
           dropdown options may not be visible before clicking
           , and the "value" field represents the option that
           will appear and be selected only after the dropdown
            is opened. The "click_point" is represented by a
           two-dimensional array indicating the absolute
           position of the click in pixels.
         - example: "action_type": "select", "click_point":
            [150, 200], "value": "Qween"
   (2) A corresponding screenshot of each operation on the
       current page. The "click_point" position of current
       action is marked with a semi-transparent red dot in
       the image.
3. The current action to be evaluated and the corresponding
   screenshot. Please note that you only need to evaluate
   the current Action (just one step within the complete
   operation sequence).

## Evaluation Criteria:
Focus on whether the action taken at the current step
   efficiently promotes task completion, not just its
   relevance to the current page:
   Level 1: The action is not the optimal choice for
       completing the task at this moment, which may lead to
       deviations from the task flow. For example:
       (1) Clicking the wrong element.
       (2) Typing incorrect or irrelevant text.
       (3) Selecting an incorrect dropdown option.
   Level 2: The action is the optimal and correct choice for
       completing the task at this moment. For example:
       (1) Clicking the correct button or link to proceed.
       (2) Typing the correct text into the appropriate field.
       (3) Selecting the correct dropdown option.

## Output requirements:
- Format: {"rating": int, "explanation": str}. Do not include
   any additional characters beyond this format.
- The "rating" field should be 1 or 2, indicating the
   evaluation level. The "explanation" field should explain
```

```
      the reasoning for this rating, without referencing any
      operations after the current step (future actions are
      unknown).

## Example Input:
Task Requirements: Search for "laptop" on an e-commerce
   website.
Action and Screenshot:
step 0: "action_type": "click", "click_point": [120, 40]
step 1: "action_type": "type", "click_point": [300, 400], "
   value": "laptop"
step 2: "action_type": "click", "click_point": [350, 400]
Current Action(to be evaluated):
step 1: "action_type": "type", "click_point": [300, 400], "
   value": "laptop"

## Example Output:
{"rating": 2, "explanation": "The action of typing 'laptop'
   into the search field is the correct and optimal choice
   for completing the task of searching for a laptop on an e
   -commerce website. This action directly contributes to
   achieving the task goal."}

Task Requirements: {}
Action and ScreenShot: {}
Current Action:
{}
```

**Prompt of MM-Mind2Web critic input**

```
As an expert in web interaction and reinforcement learning,
   you will receive textual descriptions of history
   interactions for a given web task. You need to evaluate
   the current action, similar to what a value function does
    in reinforcement learning. Detailed criteria and
   standards are given below.

## Explanation of the input content:
1. Task: Brief description of the current web task, such as "
   Search for a product on an e-commerce website".
2. Description of History operation
   Contains 3 types of web actions. Specific fields and their
       meanings are as follows:
   [1] CLICK: Click on a web element at a specific position.
      The "click_point" is represented by a two-dimensional
      array indicating the absolute position of the click in
       pixels, such as [100, 150].
      - example: "action_type": "click", "click_point": [100,
          150]
   [2] TYPE: Click and input text into a field at a specific
      position. The "click_point" is represented by a two-
      dimensional array indicating the absolute position of
      the click in pixels.
      - example: "action_type": "type", "click_point": [200,
          300], "value": "search term"
```

```
    [3] SELECT: Click at a specific position to open a
        dropdown menu, then select an option. Note: The
        dropdown options may not be visible before clicking,
        and the "value" field represents the option that will
        appear and be selected only after the dropdown is
        opened. The "click_point" is represented by a two-
        dimensional array indicating the absolute position of
        the click in pixels.
        - example: "action_type": "select", "click_point":
            [150, 200], "value": "Qween"
3. A corresponding screenshot of each operation on the
    current page. The "click_point" position of current
    action is marked with a semi-transparent red dot in the
    image.

## Evaluation Criteria:
Here are the detailed descriptions of the two levels.
    Attention needs to be paid to whether the action taken
    based on the current screenshot promotes efficient task
    execution, rather than the relevance of the content shown
     in the current screenshot to the task:
    Level 1: The action is not the optimal choice for
        completing the task at this moment, which may lead to
        deviations from the task flow. For example:
        (1) Clicking the wrong element.
        (2) Typing incorrect or irrelevant text.
        (3) Selecting an incorrect dropdown option.
    Level 2: The action is the optimal and correct choice for
        completing the task at this moment. For example:
        (1) Clicking the correct button or link to proceed.
        (2) Typing the correct text into the appropriate field.
        (3) Selecting the correct dropdown option.

## Output requirements: 1 or 2 (INT)

## Example Input:
Task Requirements: Search for "laptop" on an e-commerce
    website.
Previous Action:
step 0: "action_type": "click", "click_point": [120, 40]
step 1: "action_type": "type", "click_point": [300, 400], "
    value": "laptop"
Current Action and Screenshot:
step 2: "action_type": "click", "click_point": [350, 400]

## Example Output:
2

Task Requirements: {}
Previous Action:
{}
Current Action and Screenshot:
<image>
{}
```

**Prompt of AITW GPT-4o get negative samples input**

As an expert in the field of GUI and negative sample data
    constructor, you need to generate a new negative sample
    of the current action based on historical screenshots and
     corresponding action descriptions, task description, and
     the original current action. Detailed criteria and
    standards are given below.

## Explanation of the input content:
1. Task: Brief description of the current GUI task, such as
    implementing the "Get Hong Kong hotel prices" task in
    Android GUI.
2. History operation description and corresponding screenshot
     sequence for the task
    (1) Text description of operations: Contains 11 types of
        GUI operations. Specific fields and their meanings are
         as follows:
        [1] CLICK: Click on a specific position on the screen.
            If it is a link or software, it will enter; if it
            is text, it will be selected. The "click_point" is
            represented by a two-dimensional array indicating
            the position of the click, relative to the top-left
             corner of the screenshot and within a range from
            0.0 to 1.0.
            - example: "action_type": "CLICK", "click_point":
              [0.5, 0.5]
        [2] TYPE: An action type that sends text. Note that
            this simply sends text and does not perform any
            clicks for element focus or enter presses for
            submitting text.
            - example: "action_type": "TYPE", "typed_text": "
              capital of England"
        [3] PRESS_BACK: Return to the previous page. Usually
            the previous webpage.
            - example: "action_type": "PRESS_BACK"
        [4] PRESS_HOME: Return to the system home page. Use
            this action to return to the home screen when the
            current screen is not the desired one, so you can
            reselect the program you need to enter.
            - example: "action_type": "PRESS_HOME"
        [5] PRESS_ENTER: Press the enter key to execute a step.
             Generally, after confirming the input text, use
            this action to start the search.
            - example: "action_type": "PRESS_ENTER"
        [6] STATUS_TASK_COMPLETE: An action used to indicate
            that the desired task has been completed and resets
             the environment. This action should also be used
            if the task is already completed and there is
            nothing more to do. For example, the task is to
            turn on the Wi-Fi when it is already on.
            - example: "action_type": "STATUS_TASK_COMPLETE"
        [7] STATUS_TASK_IMPOSSIBLE: An action used to indicate
            that the desired task is impossible to complete and
             resets the environment. This can result from
            various reasons including UI changes, Android
            version differences, etc.
            - example: "action_type": "STATUS_TASK_IMPOSSIBLE"
        [8] SCROLL_DOWN: Scroll down.

```
            - example: "action_type": "SCROLL_DOWN"
        [9] SCROLL_UP: Scroll up.
            - example: "action_type": "SCROLL_UP"
        [10] SCROLL_LEFT: Scroll left.
            - example: "action_type": "SCROLL_LEFT"
        [11] SCROLL_RIGHT: Scroll right.
            - example: "action_type": "SCROLL_RIGHT"
    (2) Corresponding screenshot before each operation. If the
        operation is of the "CLICK" type, the click position
        is marked with a red dot in the image.
3. The positive current action and the corresponding
   screenshot.

## Criteria for generating negative samples:
The given input is a positive current action that meets the
   Level 2 standard below. To conduct data augmentation, we
   need to generate its corresponding negative current
   action, i.e., the action described below as level 1.
   Level 1: The action is not the optimal choice for
       completing the task at this moment, which may lead to
       deviations from the task flow. For example:
   (1) Incorrect text input.
   (2) Clicking a button that might lead to an
       advertisement.
   (3) Announcing the task's success when it has not
       actually been achieved.
   Level 2: The action is the optimal and correct choice for
       completing the task at this moment. For example:
   (1) When showing task completion, the displayed content
        can fully achieve it.
   (2) When entering an unrelated interface, you can
        return to the main screen by executing "PRESS_HOME
        ."
   (3) Selecting the most correct entry point to complete
        the current task.

## Output requirements:
- Format: {"action_desc": dict, "explanation": str}. Do not
   include any additional characters beyond this format
- The "action_desc" field needs to provide the fields
   involved in the newly generated negative sample action
   according to the text description given above. The "
   explanation" field needs to explain the logic for giving
   this new negative sample.

## Example Input:
Task Requirements: What is the capital of England?
Previous Action and ScreenShot:
step 0: "action_type": "CLICK", "click_point": "[0.524,
   0.06]"
step 1: "action_type": "TYPE", "typed_text": "capital of
   England"
Origin Action:
step 2: "action_type": "PRESS_ENTER"

## Example Output 1:
{
```

```
    "action_desc": {"action_type": "STATUS_TASK_COMPLETE"}
    "explanation": "Since text about the capital of England
        has already been entered in the search box, pressing
        enter directly at this step should give the answer.
        However, if I generate an action indicating task
        completion, it will seriously deviate from the current
         task."
}

## Example Output 2:
{
    "action_desc": {"action_type": "CLICK", "click_point":
        "[0.87, 0.52]"}
    "explanation": "Since text about the capital of England
        has already been entered in the search box, pressing
        enter directly at this step should give the answer.
        However, if I generate a click on the adjacent
        advertising area, it will deviate from the task."
}

Task Requirements: {}
Previous Action and ScreenShot: {}
Origin Action: {}
```

**Prompt of MM-Mind2Web GPT-4o get negative samples input**

```
As an expert in web interaction and negative sample data
    constructor, you need to generate a new negative sample
    of the current action based on historical screenshots and
     corresponding action descriptions, task description, and
     the original current action. Detailed criteria and
    standards are given below.

## Explanation of the input content:
1. Task: Brief description of the current web task, such as "
    Search for a product on an e-commerce website".
2. History operation description and corresponding screenshot
     sequence for the task:
    (1) Text description of operations: Contains 3 types of
        web actions. Specific fields and their meanings are as
         follows:
        [1] CLICK: Click on a web element at a specific
            position. The "click_point" is represented by a two
            -dimensional array indicating the absolute position
             of the click in pixels.
            - example: "action_type": "click", "click_point":
                [100, 150]
        [2] TYPE: Click and input text into a field at a
            specific position. The "click_point" is represented
             by a two-dimensional array indicating the absolute
             position of the click in pixels.
            - example: "action_type": "type", "click_point":
                [200, 300], "value": "search term"
        [3] SELECT: Click at a specific position to open a
            dropdown menu, then select an option. Note: The
```

```
                    dropdown options may not be visible before clicking
                    , and the "value" field represents the option that
                    will appear and be selected only after the dropdown
                     is opened. The "click_point" is represented by a
                    two-dimensional array indicating the absolute
                    position of the click in pixels.
                    - example: "action_type": "select", "click_point":
                        [150, 200], "value": "Qween"
        (2) A corresponding screenshot of each operation on the
            current page. The "click_point" position of current
            action is marked with a semi-transparent red dot in
            the image.
    3. The positive current action and the corresponding
        screenshot.

    ## Criteria for generating negative samples:
    The given input is a positive current action that meets the
        Level 2 standard below. To conduct data augmentation, we
        need to generate its corresponding negative current
        action, i.e., the action described below as level 1.
        Level 1: The action is not the optimal choice for
            completing the task at this moment, which may lead to
            deviations from the task flow. For example:
        (1) Clicking the wrong element.
        (2) Typing incorrect or irrelevant text.
        (3) Selecting an incorrect dropdown option.
        Level 2: The action is the optimal and correct choice for
            completing the task at this moment. For example:
        (1) Clicking the correct button or link to proceed.
        (2) Typing the correct text into the appropriate field.
        (3) Selecting the correct dropdown option.

    ## Output requirements:
    - Format: {"action_desc": dict, "explanation": str}. Do not
        include any additional characters beyond this format.
    - The "action_desc" field needs to provide the fields
        involved in the newly generated negative sample action
        according to the text description given above. The "
        explanation" field needs to explain the logic for giving
        this new negative sample.

    ## Example Input:
    Task Requirements: Search for "laptop" on an e-commerce
        website.
    Previous Action and Screenshot:
    step 0: "action_type": "click", "click_point": [120, 40]
    step 1: "action_type": "type", "click_point": [300, 400], "
        value": "laptop"
    Origin Action:
    step 2: "action_type": "click", "click_point": [350, 400]

    ## Example Output 1:
    {
        "action_desc": {"action_type": "click", "click_point":
            [900, 100]},
        "explanation": "Instead of clicking the search button to
            submit the query, clicking a random area on the page
```

```
            will not help complete the search task and may deviate
             from the task flow."
    }

    ## Example Output 2:
    {
        "action_desc": {"action_type": "type", "click_point":
            [300, 400], "value": "asdfgh"},
        "explanation": "Typing irrelevant text into the search
            field instead of the correct query will not help
            achieve the task goal."
    }

    Task Requirements: {}
    Previous Action and Screenshot: {}
    Origin Action: {}
```

## K  CASE STUDY

Here we randomly sample cases (Figure 6, 7, 8, 9), from our test experiments to show the difference between our method and baselines.

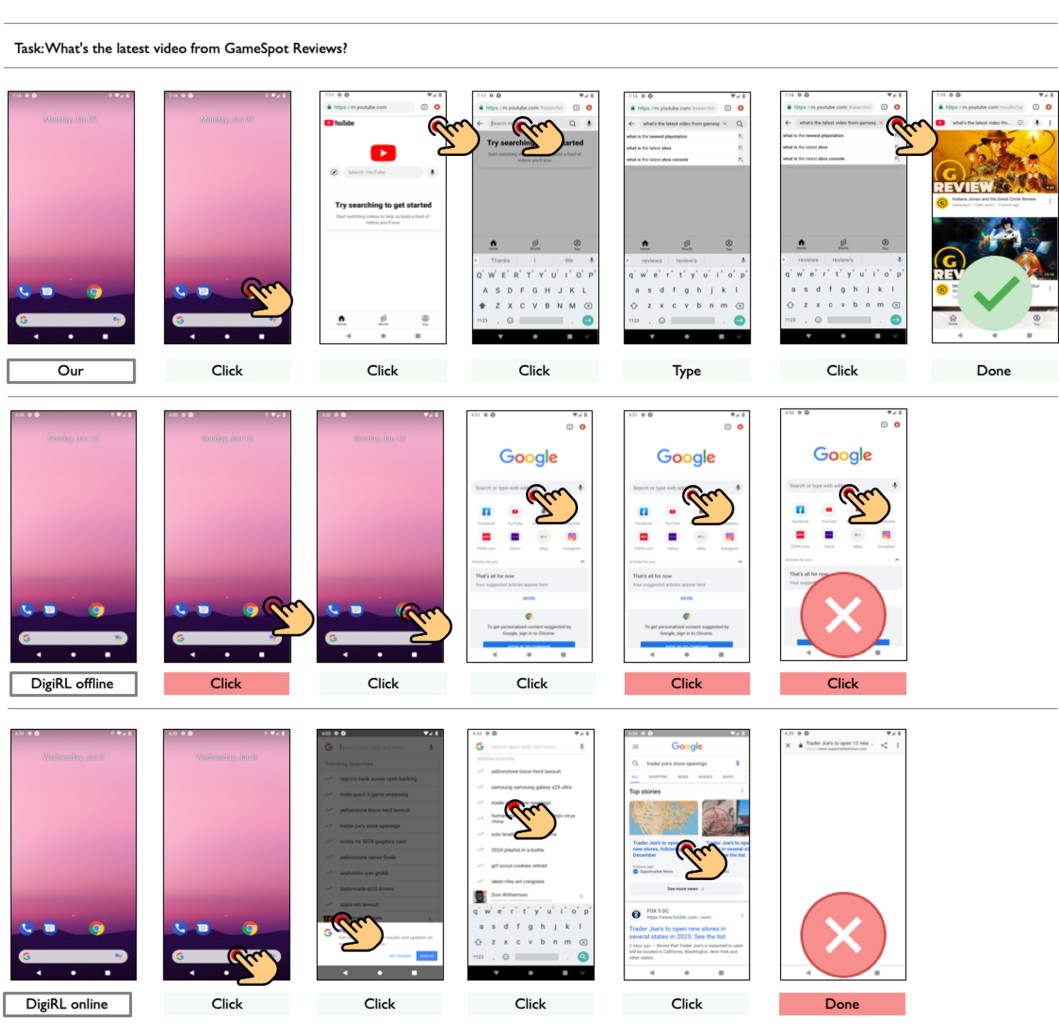

Figure 6: Case study of Ours, DigiRL offline and DigiRL online on the task: what's the latest video from GameSpot reviews?

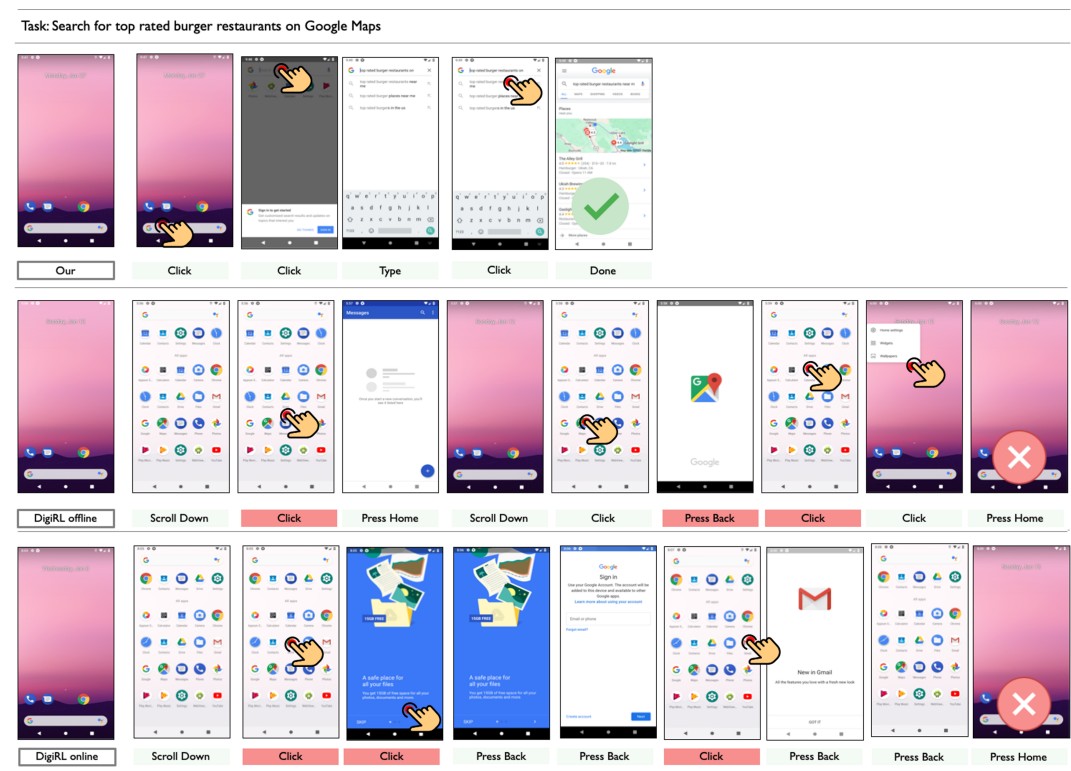

Figure 7: Case study of Ours, DigiRL offline and DigiRL online on the task: search for top rated burger restaurants on Google Maps.

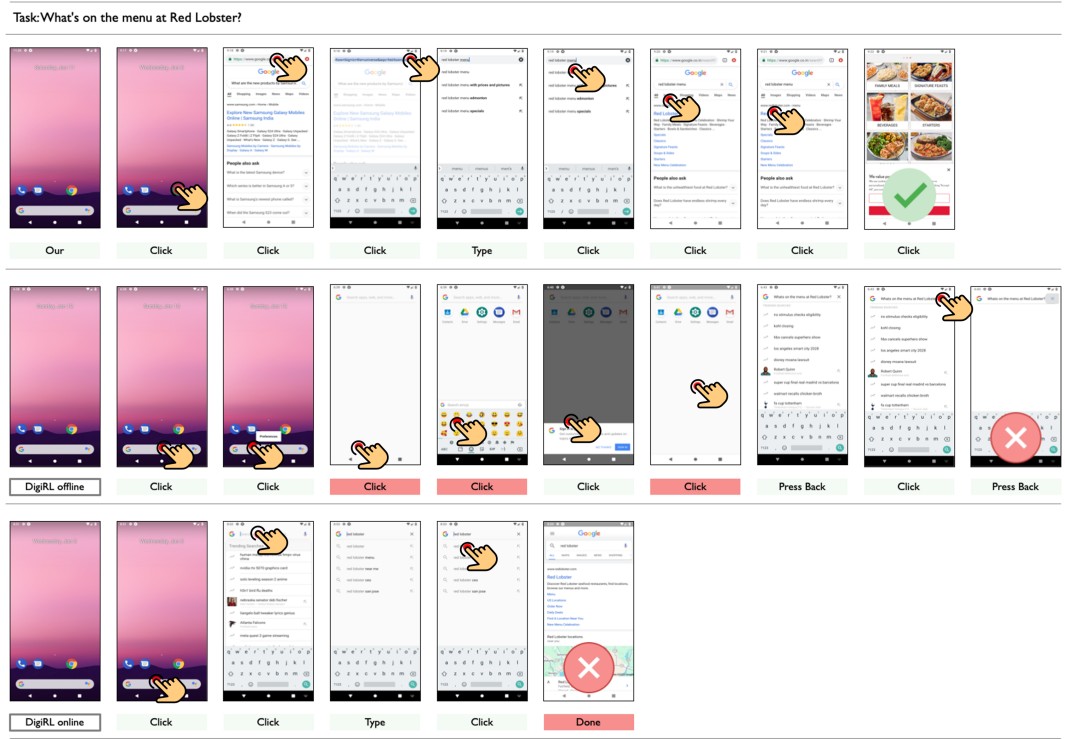

Figure 8: Case study of Ours, DigiRL offline and DigiRL online on the task: what's on the menu at Red Lobster?

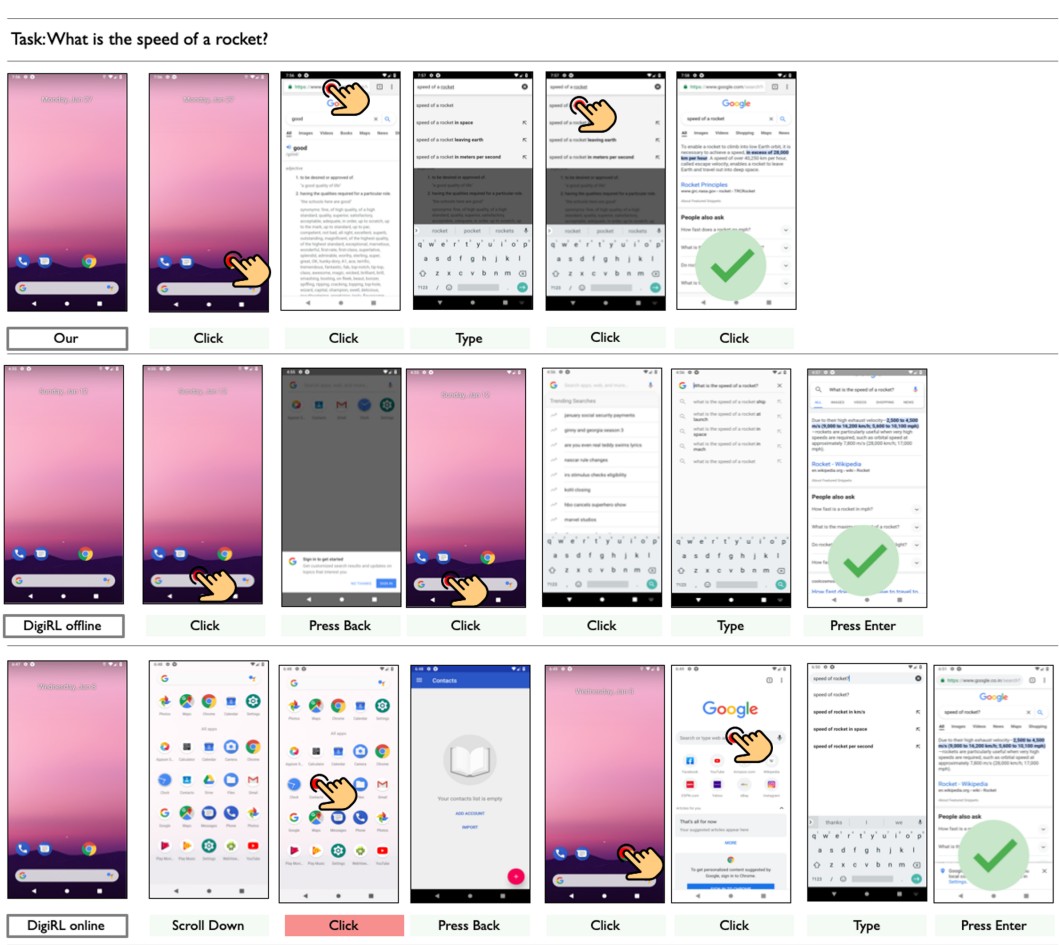

Figure 9: Case study of Ours, DigiRL offline and DigiRL online on the task: what is the speed of a rocket?

