# OpenReview forum: "VEM: Environment-Free Exploration for Training GUI Agent with Value Environment Model"
_ICLR.cc/2026/Conference — ICLR 2026 Conference Withdrawn Submission_

### Official Review · Reviewer_99m8 · 2025-10-28

**Soundness:** 3
**Presentation:** 2
**Contribution:** 2
**Rating:** 4
**Confidence:** 4

**Summary:**

This paper introduces **VEM (Value Environment Model)**, an environment-free reinforcement learning framework for training GUI agents without real environment interaction. VEM predicts state–action values directly from offline data, using GPT-4o–generated semantic annotations to guide policy learning through a frozen value model. This approach decouples value estimation from policy optimization, enabling stable, layout-agnostic training. Experiments on **Android-in-the-Wild** and **Multimodal-Mind2Web** benchmarks show that VEM achieves state-of-the-art performance comparable to environment-based methods, while drastically reducing interaction costs and improving generalization across diverse GUI environments.

**Strengths:**

The paper introduces a well-motivated framework for environment-free reinforcement learning that decouples value estimation from policy optimization through a Value Environment Model (VEM). This approach is original in enabling semantic, layout-agnostic reasoning without online interaction, offering both theoretical grounding and strong empirical results. Its demonstrated efficiency, generalization across GUI domains, and potential to scale practical GUI automation.

**Weaknesses:**

**1. Dependence on GPT-4o Annotations**
The VEM relies heavily on GPT-4o-generated annotations to train the value model. Since the accuracy of these annotations directly affects performance, the reliability and noise level of such labels remain unclear.

**2.Limited Novelty in Policy Learning**
The proposed policy learning approach follows a standard value-guided policy gradient framework similar to prior works such as RLHF and Q-function–based optimization. The main novelty lies only in the VEM pretraining process, which may be seen as an incremental extension rather than a fundamentally new policy learning method.

**3.Marginal Performance Gains**
The improvement over strong baselines like DigiQ or DigiRL is relatively small (around 3–4% on AITW) while requiring additional GPT-based annotation data. This modest gain may not fully justify the additional annotation cost or training complexity introduced by VEM.

**4.Comparison with Expert SFT Data**
The paper lacks direct experiments comparing VEM-trained policies with models fine-tuned purely on high-quality expert-labeled data.

**Questions:**

see Weaknesses.

---

### Official Review · Reviewer_cV5D · 2025-11-01

**Soundness:** 2
**Presentation:** 3
**Contribution:** 2
**Rating:** 4
**Confidence:** 4

**Summary:**

The paper presents VEM (Value Environment Model) , an environment-free RL framework for training GUI agents without requiring live environment interactions. The approach first uses GPT-4o to label offline data by assessing whether each action contributes to task completion, thereby training a value model that estimates action utility. A policy model is then optimized to maximize the predicted value within the offline dataset, enabling effective learning without direct environmental feedback.

**Strengths:**

1. The paper is easy to follow.
2. The paper introduces a novel framework by training a value model to avoid online interaction for policy improvement.

**Weaknesses:**

1. My greatest concern lies in the scalability of this method. The approach relies on data curation for value learning, where GPT-4o is used for labeling. While this may be effective in simple environments, the limitations of the model make it challenging to provide accurate value estimations in more complex or long-horizon tasks and scenarios.
2. The method should include a comparison of data efficiency with existing online learning methods.
3. The authors state that GPT-4o is used both for data creation and evaluation in Android environments. This dual usage introduces a serious concern regarding the reliability and objectivity of the results.

**Questions:**

1. Why is the reward designed as 1 for suboptimal actions and 2 for optimal actions?
2. Could we directly use the value function for planning by rolling out multiple actions and selecting the one with the highest predicted value?
3. What is the difference between learning a process reward model and learning this type of value model?
4. How reliable is the data labeled by GPT-4o? Can the labeling quality be trusted and extended to more complex scenarios?

---

### Official Review · Reviewer_adsw · 2025-11-01

**Soundness:** 1
**Presentation:** 2
**Contribution:** 2
**Rating:** 2
**Confidence:** 3

**Summary:**

The paper proposes VEM (Value Environment Model), an environment-free framework for training GUI agents. The approach has two stages: first, GPT-4o is used to annotate state-action pairs from offline trajectories with binary labels indicating whether each action contributes to task completion. A VLM is then finetuned on these labeled pairs to serve as the "value environment model". Then the policy is optimized to maximize the predicted values from VEM, enabling the agent to learn generalizable GUI manipulation policies without interacting with the real environment. Experiments on the Android-in-the-Wild benchmark show that VEM achieves competitive performance compared to environment-based methods.

**Strengths:**

- The proposed method can avoid costly online interactions by removing the need for environment rollouts.
- The method demonstrates good sample efficiency, achieving promising results with relatively few trajectories.

**Weaknesses:**

- The paper has problems of terminological imprecision. The VEM model $Q_\theta(s,a)$ is trained as a binary discriminator rather than a real Q function, and does not rely on discounted returns, TD targets, or Monte Carlo estimates. As such, referring to it as a Q-function is misleading. In my opinion it is better described as a process reward model.
- Because the policy is optimized directly on the frozen discriminator’s output, the agent’s objective can be easily exploited. This setup leads to heavy out-of-distribution (OOD) evaluation, where the VEM’s extrapolation can be unreliable. The paper lacks explicit mechanisms such as conservatism or behavior regularization to mitigate this issue.
- Since the definition of VEM is problematic, the theoretical assumptions regarding the bounded error between the learned discriminator and the true $Q^*$ are not well-justified, as the model only fits binary “progress” labels and neither models dynamics nor performs Bellman backups.
- The experiments focus primarily on AITW with limited evidence of generalization to other GUI domains.
- Several minor writing and formatting issues remain (e.g., line 202 and 214 should use citep; line 259: the en dash in “actor–critic” should be replaced with a standard hyphen).

**Questions:**

- Why is MSE used to regress binary labels instead of BCE or log-likelihood loss?
- What is the rationale for naming the VEM as $Q_\theta$? Would it be more appropriate to refer to it as a reward or process model, and to adjust the theoretical framing accordingly? Also, since the method does not predict next-state transitions or environment dynamics, why is it termed an environment model?
- The labeling process depends on a closed-source LLM (GPT-4o). How does performance change if an open-source model is used for annotation instead?
- For long-horizon tasks, how does VEM mitigate potential short-sightedness or myopic bias arising from binary labeling?

---

### Official Review · Reviewer_pB2V · 2025-11-01

**Soundness:** 2
**Presentation:** 3
**Contribution:** 2
**Rating:** 4
**Confidence:** 3

**Summary:**

This paper proposes VEM, an environment-free reinforcement learning framework for training GUI agents, aiming to avoid costly real-time environment interactions. The method works by decoupling value estimation from policy optimization. This paper first pre-trains a VEM using an offline dataset where step-level values (i.e., optimal/suboptimal actions) labeled by GPT-4o and then optimizes it with a policy gradient method to maximize the action values predicted by the VEM. Experiments show that VEM achieves competitive performance on the AITW and MM-Mind2Web benchmarks against both environment-free and environment-based baselines.

**Strengths:**

1. This paper tackles a highly practical bottleneck in GUI agent training: the high cost of real-time environment interaction.
2. The paper is well-written and clearly organized. The authors present their methodology as a logical two-stage process, and the experimental results are thoroughly analyzed.

**Weaknesses:**

1. While the method avoids GUI environment interaction, it introduces significant LLM annotation overhead . It trades environment interaction costs for LLM API costs (for both VEM supervision and negative sampling). This is essentially a "GPT-4o-in-the-loop" framework, not a pure offline method.
2. The term “Value Environment Model” might be a bit misleading. It is just a scorer, not a model of environment dynamics. The VEM operates purely as a static, frozen scorer ($Q_{\theta}$), whose sole function is to evaluate the long-term utility of a given $(s,a)$ pair. The agent does not truly interact with this model to receive new states or learn environment dynamics $P(s'|s,a)$. So calling it an environment model might not be quite accurate.
3. The VEM's generalization to OOD actions is questionable. The VEM is a static Q-function trained on a fixed dataset of Level 1 (suboptimal) and Level 2 (optimal) actions. If the policy $\pi_{\phi}$ produces a novel bad action never seen during VEM's training (which may frequently happen), the VEM's output will be unreliable. This breaks the RL loop, as the VEM cannot provide meaningful feedback for true exploration beyond its training support. The ablation results in Table 8 also back this up, showing that the “VEM-RL only” variant fails without SFT. This may suggest that the VEM can only meaningfully evaluate in-domain actions.
4. (Minior) The citation format is incorrect, e.g., uses text citations (e.g., author (year)) where parenthetical citations (e.g., (author, year)) are grammatically required.

**Questions:**

See Weaknesses

---

### Official Review · Reviewer_eqBp · 2025-11-02

**Soundness:** 2
**Presentation:** 2
**Contribution:** 2
**Rating:** 2
**Confidence:** 3

**Summary:**

This paper studies the problem of training agents for solving GUI based tasks. The authors point out that environment interactions are expensive under such settings, and prior works use sparse reward which makes online RL expensive for this problem. The paper introduces a two stage method: learning a value function based on offline data and value annotations from GPT-4o, followed by RL on the offline dataset with the given value function. The assumptions and approximations on dataset coverage, and LLM annotations are discussed and verified with real-human data. The method outperforms the baselines in terms of sample efficiency and performance across multiple GUI based tasks.

**Strengths:**

- The proposed solution is efficient as it requires fewer environment interactions as compared to the online RL baseline.
- The method is demonstrated on standard benchmarks for testing GUI-based agents, and evaluates performance across multiple baselines and tasks.
- The authors clearly discuss the approximations and limitations of the current method and also include human validation of the GPT-4o value annotations.
- The method is intuitive and easy to understand.

**Weaknesses:**

- Offline value functions for guiding online RL has been well-studied in the potential based reward shaping literature [1]. A key requirement in such a method is ensuring optimality under the sparse task-reward setting here. However, in the method introduced this seems to be not possible i.e. the GPT-4o based value functions do not guarantee that the policy is invariant to the hallucinations or inaccuracies in the labelling introduced. While the authors assume and verify that under their settings and datasets, the learned value function has high human agreement this is still a very strong assumption. I am also curious to see a discussion on combining the sparse task-success reward available with the guiding value function to ensure this invariance. Further, the literature lacks citations to more recent work in offline-RL that takes into account learning value functions offline and then learning a policy online [2].
- A baseline that directly compares these value models from GPT-4o annotations vs a standard offline-RL baseline using sparse rewards such as IQL[3], Cal-QL.
- Another baseline that is not present is something that directly does SFT on the successful trajectories from the offline dataset. Further, as according to the experiments GPT-4o is very good at labelling the actions based on the current context, if prompted with a few examples, would the zero-shot performance of the model increase. I believe this is a strong and much efficient baseline to compare against (currently the performance is zero in Table 6)
- A claim here is that web-based interactions are expensive, however, given that the entire setup is based in simulation and can be massively parallelized, it seems that the effective gains from reducing the #samples might not be that significant. On the other hand, it would also be interesting to see that given the same dataset does the offline value guidance help the agent achieve better performance.
- The introduced value function is learned via regression on the labels from GPT-4o. However, a value function has to be bellman-consistent and I believe that the current method is not ensuring that which might introduce sub-optimal behavior over the long-term (while being greedy per-step).
- A small implementation detail. Why are the labels annotated with 1,2 as compared to a binary 0/1 label?
- In the manuscript, I request the authors to fix the citation format and use \citep and \citet correctly as it makes it much easier to read and understand the context.
- Overall, the method makes strong assumptions and is missing relevant citations and baseline to more recent works in RL. I would be happy to raise my score upon clarification of my questions.



[1]. Policy invariance under reward transformations: Theory and application to reward shaping, Ng. A et. al.

[2]. Cal-QL: Calibrated Offline RL Pre-Training for Efficient Online Fine-Tuning, Nakamoto M. et. al.

[3] Implicit Q-Learning, Kostrikov, I. et al.

**Questions:**

See weaknesses above

---

### Note · Authors · 2025-11-21

I have read and agree with the venue's withdrawal policy on behalf of myself and my co-authors.